# Detection of Turbulence from Temperature, Pressure and Position Measurements Under Superpressure Balloons.

Richard Wilson [1], Clara Pitois [1], Aurélien Podglajen [2], Albert Hertzog [2], Milena Corcos [2], and Riwal Plougonven [2]

[1]LATMOS-IPSL, Sorbonne-Université, Paris, France
[2]LMD-IPSL, Ecole polytechnique, Palaiseau, France

**Correspondence:** Wilson (richard.wilson@latmos.ipsl.fr)

**Abstract.** This article deals with the detection of small-scale turbulence from in-situ meteorological measurements performed under superpressure balloons (SPBs). These balloons allow long duration flights (several months) at a prerequisite height level. The dataset is gathered from the Strateole-2 probationary campaign during which eights SPBs flew in the tropical tropopause layer at around 19 and 20.5 km altitudes, from November 2019 to March 2020.

Turbulence is not directly measured by the instrument set onboard the SPBs. Nonetheless, there is a potential to derive information about the occurrence of turbulence from the well-resolved in time measurements of pressure, temperature and position. It constitutes a challenge to extract that information from a measurement set that was not designed for quantifying turbulence, and the manuscript explains the methodology developed to overcome this difficulty.

It is observed that SPBs oscillate quasi-periodically around their equilibrium positions. The oscillation periods, 220 s in the

average, range from 130 to 500 s, close to, but noticeably smaller than, the Brunt-Väisälä period ($\sim 300$ s). The amplitude of these vertical motions is $\sim \pm 15$ m, inducing large fluctuations in all quantities, whether measured (pressure, temperature, positions) of inferred (density, potential temperature). The relationships between the changes in these quantities and vertical displacements of the balloons are used to infer properties of the flow in which the SPBs drift.

In case of active turbulence, the vertical stratification as well as the wind shear are likely to be reduced by mixing. Hence, the

increments of potential temperature, $\delta\theta$, and of the vertical displacements of the balloon, $\delta z_B$, are expected to be uncorrelated since $\partial\theta/\partial z \to 0$. Also, the local Richardson number is expected to be less than $\sim 0.25$. Several binary indexes (true or false) to describe the state of the flow, laminar or turbulent, are evaluated. They are based either on correlations between $\delta\theta$ and $\delta z_B$, or on estimations of the local Richardson number.

Correlation coefficients are computed and compared, by using different measures of $\delta\theta$ and $\delta z_B$ and by estimating either the

Pearson or the Spearman coefficients. Turbulence indexes based on a null-correlation are built using a randomisation test to check whether these correlations are significantly non-positive. It is also shown that a linear regression between the increments of a quantity and the increments of vertical displacements allows to estimate the vertical gradient of this quantity. Least square fit and Theil-Sen fit are used to estimate time series of vertical gradients $\partial T/\partial z$, $\partial u/\partial z$, $\partial v/\partial z$. Related quantities such as the Brunt-Väisälä frequency, or the local Richardson number, $Ri$, are inferred, allowing to establish turbulence indexes from $Ri$.

These different indexes, based on independent measurements and on various methods, correlations or linear regressions, are

found to be consistent: they differ for less than 3% of the cases. The flow is observed to be turbulent for about 5% of the time, with strong inhomogeneities along the longitude.

# 1  Introduction

The vertical transport of heat, momentum, and minor constituents in the tropical Upper Troposphere-Lower Stratosphere
(UTLS) is an important issue since this region is recognized as the gateway of tropospheric air into the stratosphere (Fueglistaler et al., 2009). Above 15 km altitude, this vertical transport is believed to mainly result either from the mean tropical upwelling associated with the Brewer–Dobson circulation or from the small scale turbulence. The relative contribution of turbulent mixing to this vertical transport is highly uncertain, partly owing to lack of observations. Turbulence observations in the tropical UTLS are indeed sparse: they mostly come from two large VHF radars, from relatively few radiosondes, or from research
aircraft.

From measurements of the Equatorial Atmospheric Radar (EAR) located in West Sumatra, Indonesia (0.20°S, 100.32°E), Fujiwara et al. (2003) observed intermittent turbulence near the tropical tropopause with significant enhancements, a factor of 5 in the turbulent kinetic energy (TKE), lasting several days. Such enhancements in the TKE are believed to result from the breaking of large scale Kelvin waves. By using the same EAR data set, Yamamoto et al. (2003) showed that eastward vertical
wind shear around the equatorial tropopause frequently generates turbulence through Kelvin-Helmholtz (KH) instabilities. Mega et al. (2010) presented detailed structures of KH instabilities in the equatorial UTLS from both high-resolution EAR measurements (by using an interferometric imaging method), and radiosondes. From the VHF radar of Gadanki, India (13.5°N, 79.2°E) Satheesan and Murthy (2002) and Satheesan and Krishna Murthy (2004) described turbulence characteristics in the tropical UTLS. These authors estimated turbulent kinetic energy (TKE) and TKE dissipation rates from several methods.
Interestingly, they did not observed a clear variability with altitude of the turbulence intensity within the UTLS.

Sunilkumar et al. (2015) and Muhsin et al. (2016) presented characteristics of turbulence in the tropical UTLS from GPS-radiosondes observations obtained during more than three years at two stations located in the Indian Peninsula, Trivandrum (8.5°N, 76.9°E) and Gadanki (13.5°N, 79.2°E). The turbulent layers are detected by the Thorpe's analysis (Thorpe, 1977) following the procedure proposed by Wilson et al. (2010, 2011, 2013). The statistics of various turbulence parameters are
described, as well as Brunt-Väisälä frequencies and vertical shears, this for the convective troposphere and for the UTLS. The parameters describing the turbulence are either directly measured, such as the Thorpe lengths (an outer scale of turbulence) and the frequency of appearance of unstable layers, or are inferred on the basis of physical assumptions, like the TKE dissipation rates and the eddy diffusivity. Muhsin et al. (2020) extended the two previous studies by analyzing the soundings of six stations of South-India, adding stations of Cochin (10°N, 76.3°E), Coimbatore (10.9°N, 76.9°E), Goa (15.5°N, 73.8°E) and Hyderabad
(17.5°N, 78.6°E) to the two previously mentioned. These data were acquired during four years, from August 2013 to December 2017. All these studies based on radiosondes measurements consistently show that the probability of occurrence of instability, either estimated from the gradient Richardson number $Ri$ or from the squared Brunt-Väisälä frequency $N^2$, is decreasing with altitude above 15 km altitude, i.e. in the UTLS. They found that the probability of occurrence for $Ri$ to be less than 0.25 is

between 0 and 5% in the height range $18-25$ km, and that the probability of occurrence of unstable region ($N^2 < 0$) is quasi null. Interestingly, Muhsin et al. (2016) did not observe a clear diurnal cycle in the tropical UTLS.

Many studies of turbulence in the free atmosphere are based on measurements from research aircraft (see Dörnbrack et al., 2022, and references therein) but, to our knowledge, only one study involves the tropical UTLS. Podglajen et al. (2017) used high-resolution (20 Hz) airborne measurements to study the occurrence and properties of small-scale ($<\sim 100$ m) wind and temperature fluctuations in the tropical UTLS over the Pacific ocean. They show that wind fluctuations are very intermittent and appear to be localized in shallow layers, with $\sim 100$ m thickness typically. Furthermore, active turbulent events appear to be more frequent at relatively low altitude and near deep convection. They observe that the motions are close to 3D isotropic and that the power spectra follow a $\sim -5/3$ power-law scaling. The diffusivity induced by turbulent bursts is estimated to be in the order of $10^{-1}$ m$^2\cdot$s$^{-1}$ and decreases from the bottom to the top of the tropical UTLS.

Apart from convective regions, many observations of the free atmosphere show successions of strata, i.e. alternating layers in which the flow is turbulent, whose depth varies from a few tens to a few hundreds of meters, separated by stable static regions, i.e. regions where the flow is laminar (e.g., Luce et al., 2015; Podglajen et al., 2017; Wilson et al., 2018). The numerical simulations of Fritts et al. (2003) support such observations. Turbulent and laminar strata are expected to exist in the tropical UTLS. Since turbulence, by nature, is a dissipative process, the turbulent layers have a finite lifetime, depending on the process yielding energy at meso scale, shear instability or wave breaking. Thus, by observing the flow at a given altitude level, we expect to find an alternation of turbulent and laminar episodes.

The Strateole-2 project was set up in recent years in order to better understand the dynamics, transport, microphysics and dehydration of the tropical UTLS (Haase et al., 2018). It is an international project involving several research groups mostly in France and USA, led by Laboratoire Meteorologie dynamique (LMD) and Centre National d'Etude Spatiale (the French Space agency). The uniqueness and strength of the Strateole-2 project come from the fact that the atmospheric measurements are obtained under super pressure balloons (SPB) which can fly for several months (typically three months) at an approximately constant density level. Standard measurements performed under all the Strateole-2 SPBs, including temperature, pressure and GPS positions, allow to describe meso- and small-scale dynamic processes at the flight level, in a quasi-Lagrangian way (Hertzog and Vial, 2001; Hertzog et al., 2012; Podglajen et al., 2016; Corcos et al., 2021). The SPBs usually carry a second gondola with several configurations of instruments, allowing measurements of constituents, aerosols, cloud heights, radiation, etc.

The main purpose of this paper is to present methods allowing to determine the dynamical state of the flow, laminar or turbulent, in which the SPBs are flowing in. It is based on the estimation of the local stratification of the flow using measurements of temperature, pressure and GPS positions.

The paper is organized as follow: the used data set is presented in the second section, the methods for detecting turbulence and for estimating various indicators of the flow stability are described in the third section, some results are presented and discussed in the forth section, a concluding section summarizes the main findings of the paper.

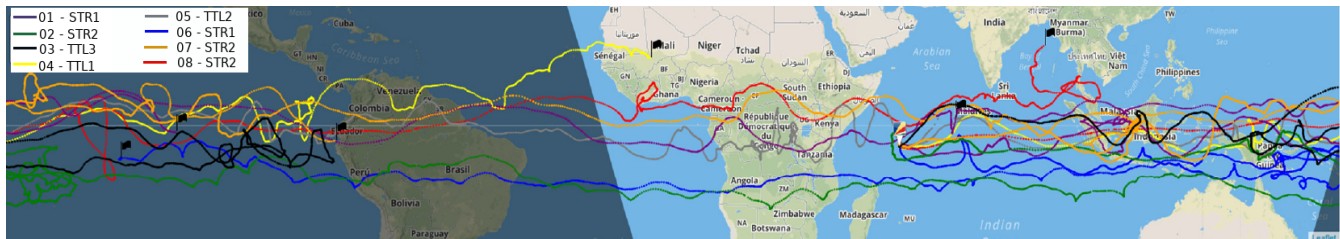

**Figure 1.** Trajectories of the eight SPBs

## 2 The data set

### 2.1 The Strateole-2 C0 campaign

The probatory Strateole-2 campaign, called Strateole-2 C0, was held from November 2019 to March 2020. During this cam-
95 paign, eight SPBs were launched from Mahé, Seychelles (4°37'S, 55°27'E), and flew eastward following the Quasi-Biennial
Oscillation (QBO) phase that prevailed in the lower stratosphere at this time (Figure 1). The main characteristics of the flights
are provided in Table 1. Four of these flights carried only the TSEN (Thermodynamics SENsors) instrument and its associated
GPS receiver, the other four carried additional scientific instruments. At first order, SPBs fly on a quasi-constant density level
(Vincent and Hertzog, 2014). TTL and STR flights were associated with different carried masses and/or balloon sizes, and
100 respectively drift at altitudes of $\sim 19$ km and $\sim 20.5$ km.

| Flight Id | Height (m) | Launch date (UT) | Duration (days) |
|---|---|---|---|
| 01_STR1 | 20652 | 2019-11-12 | 107 |
| 02_STR2 | 20502 | 2019-11-11 | 103 |
| 03_TTL3 | 19110 | 2019-11-18 | 101 |
| 04_TTL1 | 18780 | 2019-11-27 | 66 |
| 05_TTL2 | 18800 | 2019-12-05 | 79 |
| 06_STR1 | 20380 | 2019-12-06 | 57 |
| 07_STR2 | 20125 | 2019-12-06 | 83 |
| 08_STR2 | 20100 | 2019-12-07 | 77 |

**Table 1.** Main characteristics of Strateole-2 C0 flights. All superpressure balloons are equipped with TSEN sensors.

### 2.2 The TSEN measurements

In this study, we use the TSEN measurements that are performed at the balloon flight level for each flight. They respectively
consist of two temperature measurements, one with a thermistor sensor ($T_S$), the other with a thermocouple sensor ($T_C$), and a
pressure measurement ($P$). They are closely associated with GPS observations that provide the position (lon, lat, altitude asl)

of the balloon, as well as the solar zenith angle (SZA). All these measurements are acquired every 30 s, except for the pressure measurements that are acquired every 1 s.

These raw measurements enable us to compute two estimates of the balloon potential temperature ($\theta_S$ and $\theta_C$) and density ($\rho_S$ and $\rho_C$), respectively obtained with the thermistor and thermocouple temperatures. The horizontal velocity components ($u$ and $v$) are evaluated from successive GPS positions.

In the rest of the study, we will make use of the increments of measured quantities, namely the differences between consecutive measurements: $\delta T_S$ will for instance stand for the increments of the thermistor temperatures. Note that two independent estimates of the balloon vertical displacements are available: $\delta z_{\mathrm{GPS}}$ obtained with the raw GPS altitudes, and $\delta z_P$ computed with the pressure measurements and further assuming hydrostaticity.

TSEN measurements were also performed on the profiling unit of the Reeldown Aerosol Cloud Humidity and Temperature Sensors (RACHuTS) instrument (Kalnajs et al., 2021). RACHuTS notably allows to obtain vertical temperature profiles of 2 km length below the balloon by reeling down (and then up) the profiling unit at a vertical velocity of about 1 m/s. Only one TSEN temperature sensor (a thermistor) is implemented in the profiling unit. Measurements are performed with a nominal sampling rate of 1 Hz, i.e. the vertical resolution of the RACHuTS temperature profiles is $\sim$ 1 m, which we degraded to about 30 m in order (1) to improve upon the 1-m raw precision of altitude measurements and (2) to reach similar vertical resolution from TSEN and RACHuTS measurements (since the amplitude of the balloon oscillations is typically $\pm$ 15 m).

### 2.2.1 Instrumental noise

The instrumental noise of the TSEN measurements is an important characteristic that has to be taken into account to assess whether the flow is turbulent. It is assumed that this noise is a zero-mean, uncorrelated process, i.e. a white noise, contributing to the measured signal. A way to estimate the white noise level is to compute the variance of the $n$-th order increments of the time series ($n \gtrsim 6$), such a high order differentiation performing a high-pass filtering of the time series. As shown in the Appendix, the variance of the $n$-th order increments tends to the variance of the uncorrelated signal weighted by the sum of squared binomial coefficients of order $n-1$. Noise levels are here estimated on time segments of 21 samples (10 min), that is on more than $10,000$ segments for a flight lasting $\sim 100$ days. This method works well if the signal spectra exhibit a quasi-constant floor, i.e. if a white noise clearly contributes to the measured signal for frequencies smaller than the Nyquist frequency, as is frequently the case for lidar or radar signals. However, we do not observe any white noise level for the TSEN measurements acquired with a sampling period of 30 s ($T_S$, $z$, $u$) (see for instance the power spectra of the vertical displacements $\delta z$ shown in Fig 6). Therefore, the variance of the $n$-th order increments cannot be interpreted as solely due to uncorrelated noise, even if some noise is expected to contribute to this high-frequency variance. However, a rough estimate of the noise level has been evaluated from the data segments showing the smallest variances, i.e. segments for which a comparatively smaller contribution of the atmospheric signal is expected. Tables (2) and (3) display estimates of the noise levels for quantities measured or inferred, obtained with the average of the 10% smallest variances of the 6-th order increments. Such estimates, although not fully satisfying, provide however valuable information on the relative quality of the data, allowing to compare noise levels between sensors, between flights or between night and day measurements.

|  | $T_S$ (mK) | $T_C$ (mK) | P (Pa) | $\theta_S$ (mK) | $\theta_C$ (mK) |
|---|---|---|---|---|---|
| 01_STR1 | **7** | **7** | **0.1** | **17** | **17** |
|  | $5-27$ | $5-22$ | $0.08-0.12$ | $12-64$ | $13-75$ |
| 02_STR2 | **2** | **2** | **0.05** | **5** | **5** |
|  | $1-6$ | $2-6$ | $0.04-0.06$ | $3-15$ | $4-15$ |
| 03_TTL3 | **14** | **10** | **0.12** | **31** | **21** |
|  | $8-36$ | $7-23$ | $0.11-0.15$ | $19-78$ | $15-52$ |
| 04_TTL1 | **9** | **8** | **0.17** | **19** | **16** |
|  | $6-31$ | $6-23$ | $0.15-0.18$ | $13-67$ | $12-50$ |
| 05_TTL2 | **3** | **3** | **0.06** | **6** | **6** |
|  | $2-7$ | $2-9$ | $0.05-0.07$ | $4-16$ | $4-20$ |
| 06_STR1 | **6** | **6** | **0.08** | **13** | **15** |
|  | $4-26$ | $5-25$ | $0.07-0.09$ | $9-62$ | $11-60$ |
| 07_STR2 | **5** | **7** | **0.09** | **12** | **16** |
|  | $4-18$ | $5-23$ | $0.08-0.12$ | $9-42$ | $12-54$ |
| 08_STR2 | **3** | **3** | **0.06** | **7** | **7** |
|  | $2-8$ | $2-8$ | $0.05-0.07$ | $5-20$ | $6-20$ |

**Table 2.** Estimated noise (standard deviation) of temperature, pressure and dry potential temperature. For each flight, the first row indicates the flight-mean values (bold). The second row respectively indicates the nighttime and daytime average values.

The noise levels of the two temperature sensors, thermocouple and thermistor, are quasi identical. We note that the noise level of temperature measurements is a factor of three to five larger during daytime than during nighttime for all flights. Such an increase in temperature noise during day very likely results from the random passage of the sensors in the wake of their electrical wires or mechanical support. These devices, which are significantly thicker than the sensors themselves, are heated by the solar radiation during day, and are consequently warmer than the air or even the sensor temperature. We also observe that flight 03_TTL3 temperature measurements are the noisiest (14 mK vs $2-9$ mK). Unlike on other flights, where temperature sensors are hanging at the very bottom of the flight train as far away as possible from large elements (e.g. the gondolas), they were located within the flight train on flight 03_TTL3. They were thus more prone to being affected by the warm wake of other devices in the flight train during day.

The noise estimates on the pressure measurements, as well as on the vertical positions (both from pressure differences or GPS) are slightly larger during daytime than during nighttime. The reason for this slight increase in the noise level is probably not the consequence of an increase in instrumental noise. It may rather be the consequence of the greater amplitudes of the balloon oscillations during day, which would impact the energy density of $P$ and $z$ at high frequency. The noise levels of the horizontal wind components estimated from GPS positions is very small, less than 1 cm/s. They do not show any night/day difference.

| | $z_P$ (m) | $z_{GPS}$ (m) | u (cm/s) | v (cm/s) |
|---|---|---|---|---|
| 01_STR1 | **0.11** | **0.32** | **0.2** | **0.2** |
| | $0.10 - 0.14$ | $0.29 - 0.36$ | $0.2 - 0.3$ | $0.2 - 0.3$ |
| 02_STR2 | **0.05** | **0.10** | **0.1** | **0.1** |
| | $0.04 - 0.07$ | $0.1 - 0.120$ | $0.1 - 0.1$ | $0.1 - 0.1$ |
| 03_TTL3 | **0.11** | **0.26** | **0.2** | **0.2** |
| | $0.10 - 0.13$ | $0.24 - 0.27$ | $0.2 - 0.3$ | $0.2 - 0.3$ |
| 04_TTL1 | **0.14** | **0.32** | **0.3** | **0.2** |
| | $0.13 - 0.15$ | $0.30 - 0.33$ | $0.2 - 0.3$ | $0.2 - 0.3$ |
| 05_TTL2 | **0.05** | **0.12** | **0.1** | **0.1** |
| | $0.05 - 0.06$ | $0.12 - 0.13$ | $0.1 - 0.2$ | $0.1 - 0.1$ |
| 06_STR1 | **0.09** | **0.28** | **0.6** | **0.2** |
| | $0.08 - 0.10$ | $0.27 - 0.30$ | $0.5 - 0.7$ | $0.2 - 0.2$ |
| 07_STR2 | **0.10** | **0.27** | **0.2** | **0.2** |
| | $0.09 - 0.13$ | $0.26 - 0.29$ | $0.2 - 0.2$ | $0.2 - 0.2$ |
| 08_STR2 | **0.07** | **0.27** | **0.1** | **0.1** |
| | $0.06 - 0.08$ | $0.23 - 0.31$ | $0.1 - 0.1$ | $0.1 - 0.1$ |

**Table 3.** Same as Table 2, but for altitudes derived from the pressure and altitude observations, as well as zonal and meridional velocities.

## 2.3 Oscillating movements of the balloons around their equilibrium positions

Superpressure balloons drift with the winds, following a quasi-Lagrangian behavior useful to document the detailed evolution of a given air mass. Their displacements however differ from those of an air parcel in at least two ways: first and most importantly, the balloons follow isopycnic trajectories. The balloon envelope is almost inextensible, the balloon diameter varying by less than 1%, hence the volume is fixed as long as a superpressure is present. As the total mass of the flight train is fixed, the density of the balloon remains constant, implying an isopycnic trajectory. In contrast, air parcels follow isentropic trajectories

in the absence of diabatic forcing. A second difference comes from the existence of natural oscillations of the balloon around its equilibrium density surface (EDS), where it achieves neutral buoyancy. These oscillations are of minor importance for the study of phenomena on timescales larger than a few tens of minutes (Vincent and Hertzog, 2014). They will yet be central to the current study, because an oscillating balloon is sampling short vertical profiles of key meteorological variables. The methodology developed in the present study precisely aims at exploiting these short, 'unintended', vertical profiles to diagnose

the occurrence of turbulence.

The atmospheric density ($\rho$) at the flight level is estimated from the measurements of temperature ($T$) and pressure ($P$) by using the perfect gas law, $\rho = P/R_a T$ where $R_a$ is the ideal gas constant for dry air per unit mass. Since two independent measurements of $T$ exist, two estimates of $\rho$ can be calculated, denoted $\rho_S$ and $\rho_C$, by using respectively the thermistor or the

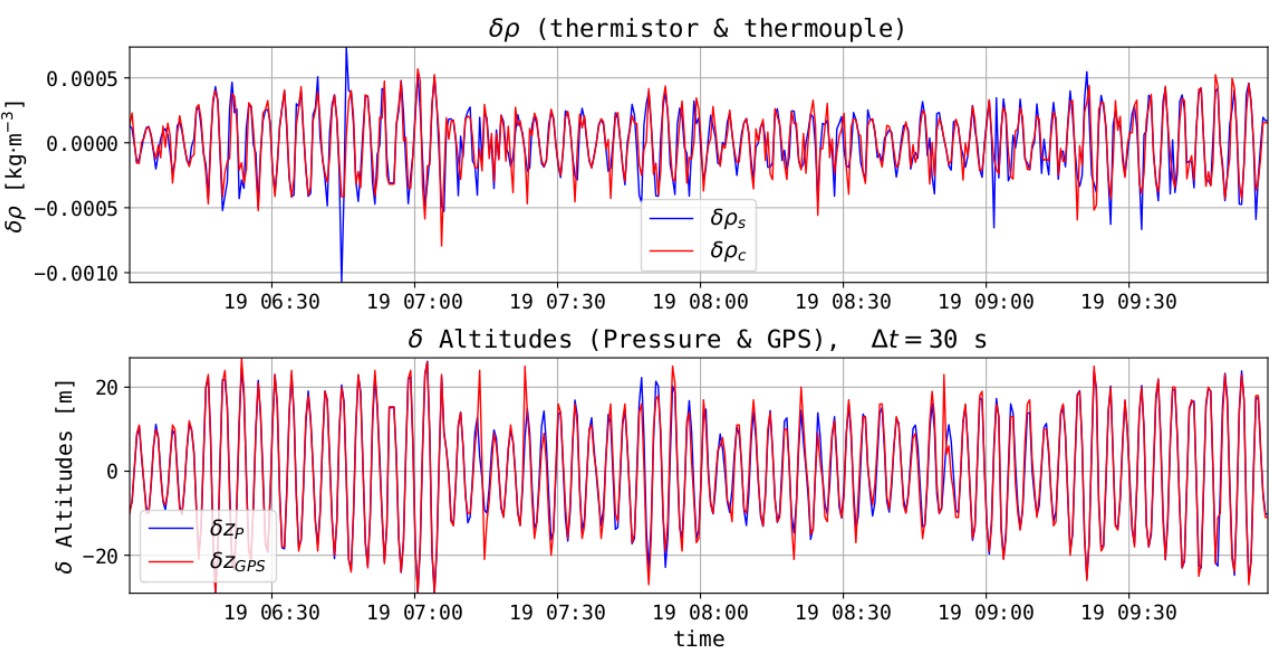

**Figure 2.** Top: density fluctuations observed on November 19th, from 06:00 to 10:00. Bottom: fluctuations in the height of the balloon for the same time interval.

thermocouple temperatures. Figure 2 displays the systematic balloon fluctuations in density about their EDS (top panel), as well as the related variations in balloon altitudes (bottom panel) during four hours of flight 01_STR1. The fluctuations in density are about $\pm 5 \times 10^{-4}$ kg·m$^{-3}$, corresponding to relative fluctuations $\sim \pm 0.5\%$. The corresponding amplitude of the balloon vertical displacements is in this case $\pm 20$ m. SPBs thus move around their EDS, exploring the atmosphere above and bellow over a few tens of meter. These oscillatory motions induce fluctuations in the measured ($P$, $T$, $z_{\mathrm{GPS}}$) and inferred quantities ($\rho$, $\theta$, $z_P$). We shall use these variations to describe some properties of the flow by making the hypothesis that the observed variability of a quantity depends on both the balloon vertical displacement and the local vertical gradient of this quantities.

### 2.3.1 Periods and amplitudes of the vertical oscillations

Three angular frequencies, close to each other, need first to be distinguished: the Brunt-Väisälä frequency ($N$), the neutral buoyant oscillation frequency ($\omega_{\mathrm{NBO}}$), and the observed frequency of SPB oscillations ($\omega_B$). The frequency at which a spherical balloon with constant volume oscillates about its EDS, the neutral buoyant oscillation (NBO) frequency, reads (Hanna and Hoecker, 1971; Nastrom, 1980; Vincent and Hertzog, 2014):

$$\omega_{\mathrm{NBO}}^2 = \frac{4\pi^2}{t_{\mathrm{NBO}}^2} = \frac{2g}{3T}\left(\frac{\partial T}{\partial z} + \frac{g}{R_a}\right) \tag{1}$$

where $t_{\text{NBO}}$ is the NBO period, and $g$ is the acceleration of gravity. The squared Brunt-Väisälä (BV) frequency, $N^2 = g/T(\partial T/\partial z + g/c_p)$ can be expressed as a function of $\omega_{\text{NBO}}^2$:

$$N^2 = \frac{3}{2}\omega_{\text{NBO}}^2 - \frac{g}{\gamma H} = \frac{3}{2}\omega_{\text{NBO}}^2 - \frac{5}{7}\frac{g}{H} \tag{2}$$

where $c_p$ is the air specific heat capacity at constant pressure, $\gamma$ is the heat capacity ratio, and $H = R_a T/g$ is the atmospheric scale height.

From Eq. 2, it can be shown that $N \leq \omega_{\text{NBO}}$ as long as $\omega_{\text{NBO}}^2 \leq 10g/7H$, i.e. $t_{\text{NBO}} \geq 2.8\pi^2 H/g \approx 120$ s, or $\partial T/\partial z \leq 8g/7R_a \approx 39$ K/km. Such conditions are met very frequently in the lower stratosphere, if not always. Note that both $\omega_{\text{NBO}}$ and $N$ increase with the vertical gradient of temperature $\partial T/\partial z$.

The top panel of Figure 3 displays the observed periods of the balloon vertical oscillations ($t_B$) for the whole 01_STR1 flight. The gray and white stripes correspond to nights and days respectively, the orange curve showing the daytime- and nighttime-averaged values. The histogram of the oscillations periods is shown on the bottom left panel, the periods are observed to range from $\sim 135$ s to 800 s. No clear day/night variation is visible. The bottom right panel shows the cumulative distribution function (CDF) of the periods: the median $t_B$ value is close to 220 s, while the mean is $\sim 250$ s. Hence, $\omega_B$ is larger than $N$, as long as $N$ is smaller than $\approx 2.5 \times 10^{-2}$ rad/s, a typical value in the tropical lower stratosphere. We shall see below that $\omega_B$ is systematically larger than $N$.

Figure 4 shows the corresponding amplitudes of the observed balloon oscillations, as well as their probability and cumulative distribution function. The observed amplitudes range from $\sim 0$ to 150 m, with an average (resp. median) of 14 m (resp. 15 m) respectively. Some large amplitude oscillations (>100 m) are observed. We found that they are most often associated with depressurization events. A weak but clear day/night variability is found since the oscillations amplitudes are about 20% larger during daytime than during nighttime. Also, greater variability in altitudes is observed during daytime at high frequencies, which was reported in table 3.

## 3   Data processing

### 3.1   Methods for detecting the occurrence of turbulence from the TSEN measurements

The time scales of the turbulent fluctuations are expected to be smaller than the Brunt-Väisälä period, $t_N$. (In the following, we shall refer to "high frequencies" for frequencies larger than $N/(2\pi)$.) As $t_B$ is close, or even smaller than $t_N$, the high-frequency variability up to the Nyquist frequency (1/60 Hz) will be dramatically affected by the balloon oscillations, whichever the state of the flow, laminar or turbulent. Hence, the detection of turbulence from either the variance, or from the spectral characteristics of the TSEN measurements at high frequencies appears difficult, if not impossible.

Diagnostics on the dynamical state of the flow can therefore only be based on the local properties of atmospheric stratification, either stable or neutral/unstable. A consequence of turbulence is to restore stability from a preceding unstable state of the flow, which is achieved by locally mixing the fluid. In this case, the conservative quantities, potential temperature, specific humidity, momentum, are expected to exhibit weak horizontal and vertical variability within the turbulent layer. Note that neutral

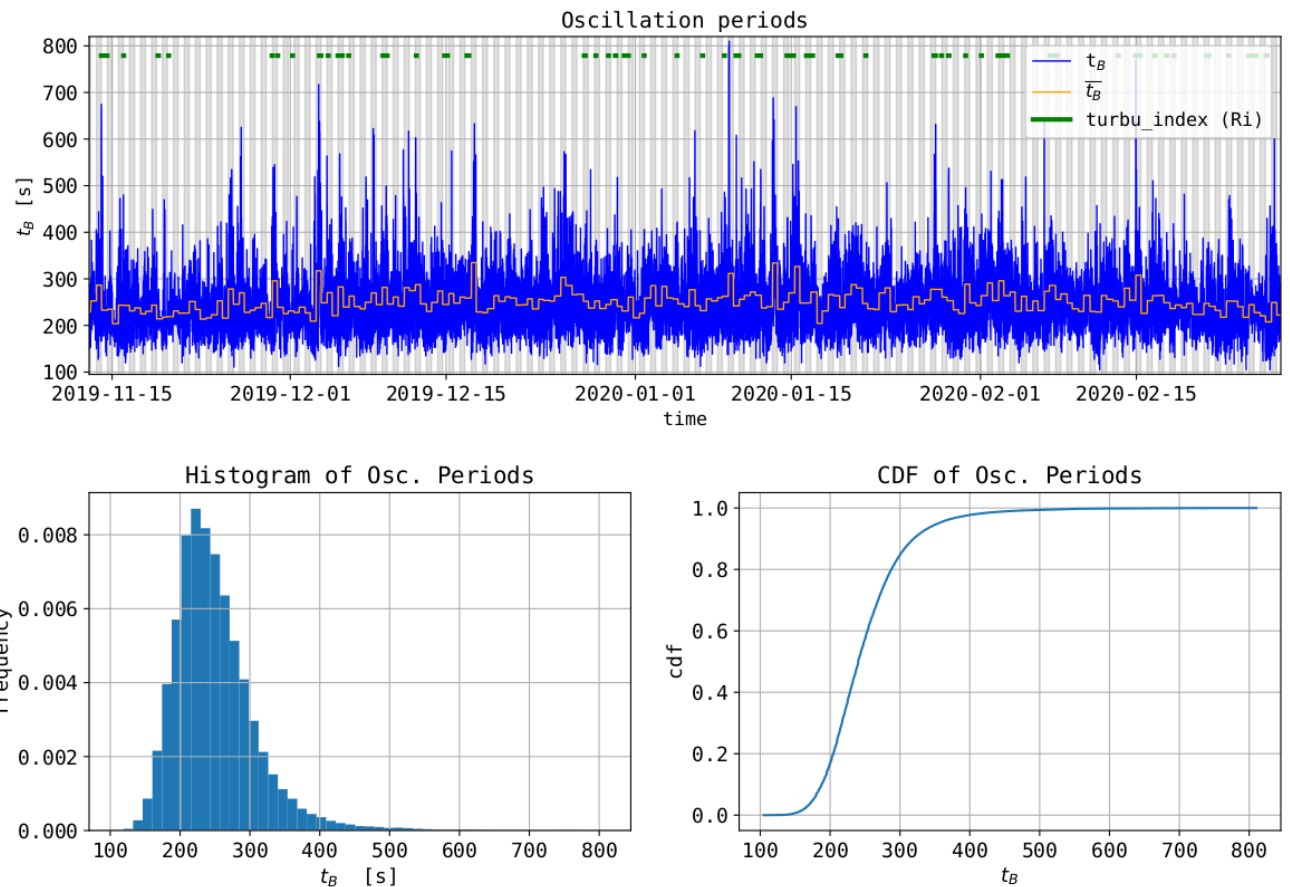

**Figure 3.** Top: estimates of the SPB oscillation periods during flight 01_STR1, the orange staircase curve showing daytime and nighttime averages. Bottom left: Histogram of the oscillation periods. Bottom right: cumulative distribution function of the oscillation periods.

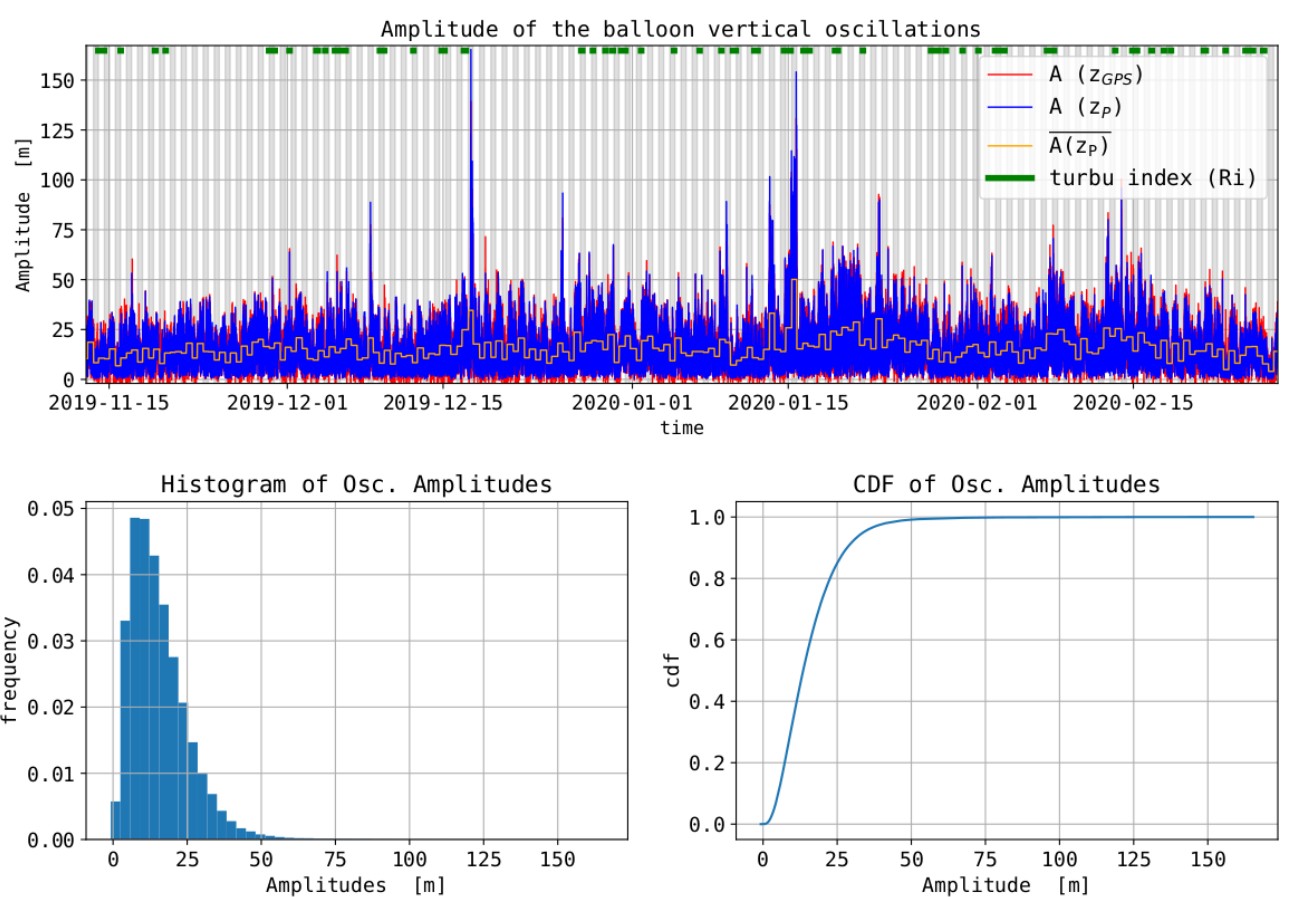

**Figure 4.** Top: Estimates of SPB oscillation amplitudes during flight 01_STR1, the orange staircase curve showing daytime and nighttime averages. Bottom left: Histogram of the oscillations amplitudes. Bottom right: CDF of the oscillations amplitudes

or even unstable stratification conditions may precede turbulence, but such conditions cannot persist and will necessarily cause turbulence. Conversely, a laminar flow is expected to exhibit significant variability of the same quantities along the vertical direction.

Numerical simulations indicate that thin layers with large temperature gradients are expected at the edges of turbulent layers (Fritts et al., 2003; Werne and Fritts, 1999). These large temperature gradients are also commonly observed from radiosonde profiles when turbulence detection is performed by the Thorpe method (e.g. Luce et al., 2002). It is clear that the sampling by a balloon drifting within an air mass, and not cutting vertically through it as a radiosonde, will not allow to identify as turbulent such temperature gradients. Only the central part of the turbulent region, in which stratification is quasi neutral, can be identified as turbulent.

A first way to diagnose that the flow is turbulent is thus to detect a null gradient in potential temperature. In such a case, the correlation between the vertical displacements ($\delta z_B$) and the corresponding potential-temperature increments ($\delta\theta$) is expected to be null. The implementation therefore consists in testing the $H_0$ hypothesis of a null correlation between $\delta z_B$ and $\delta\theta$. This method is hereafter called the correlation method. A second way to diagnose a turbulent flow is based on the estimation of the local Richardson number $Ri = N^2/S^2$, where $S^2 = (\partial u/\partial z)^2 + (\partial v/\partial z)^2$ is the squared shear. The flow is assumed to be turbulent when the Richardson number becomes less that $0.25$. This method is hereafter called the $Ri$ method.

## 3.2 Implementation of the two methods of turbulence detection

### 3.2.1 Relationship between the increments of measured quantities and the increments of vertical displacements

The two proposed methods of turbulence detection are based either on estimates of correlations (for the correlation method) or on linear regressions (for the Richardson method) between increments. The increments are simply defined as the differences between consecutive measures, separated by $\delta t = 30$ s. Such a differentiation realizes a high-pass filtering of the time series. Both the correlations and the linear regressions are computed over time periods of 1 hr, i.e. with 120 30-s observations. The choice of this number of observations enables to obtain relatively small uncertainties on the estimates of correlation coefficients and slopes, at the price of being unable to detect turbulence layers with time scale significantly shorter than 1 hr (in the frame moving with the wind).

The oscillatory motions of SPBs at frequency $\omega_B$ are expected to occur neither on isentropic surfaces nor on isopycnic surfaces. If the vertical displacement of the balloon between two different times separated by $\delta t = 30$ s is $\delta z_B = z_2 - z_1$, the change in the measured potential temperature $\delta\theta$ depends on both $\delta z_B$ and on the vertical displacement of the atmosphere $\delta\zeta_\theta$ during $\delta t$:

$$\delta\theta = \bar{\theta_z}(z_2 - z_1') = \bar{\theta_z}(\delta z_B - \delta\zeta_\theta) \tag{3}$$

$\delta\zeta_\theta = z_1' - z_1$ corresponding to the change in the height of the isentropic surface during $\delta t$ (see Fig 5). Eq. 3 must be modified for non-conservative quantities, such as temperature $T$ or horizontal velocities, $u$ and $v$, in order to take into account the change

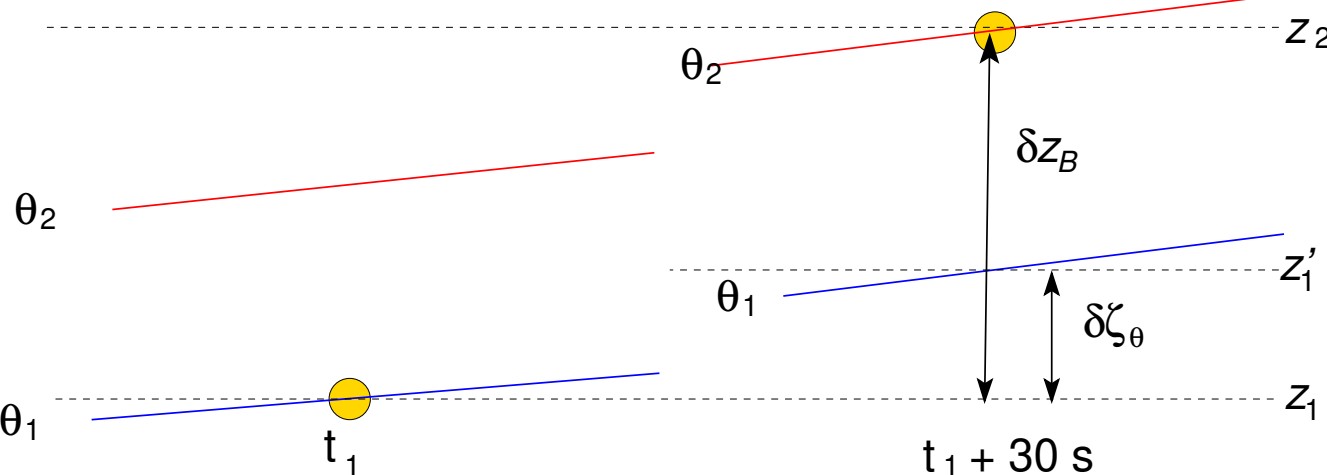

**Figure 5.** Schematic of the vertical displacement of a SPB. Between the measurement times $t_1$ and $t_1 + 30$ s, the vertical displacements de the SPB is $\delta z_B$, and the vertical displacement of the isentropic surface is $\delta \zeta_\theta$.

of the considered quantity during $\delta t$ on the isentropic surface. For instance, the temperature increment reads:

$$\delta T = \bar{T}_z \delta z_B - \left( \bar{T}_z + \frac{g}{c_p} \right) \delta \zeta_\theta \tag{4}$$

Figure 6 shows the power spectral density (PSD) of the vertical displacements ($\delta z_B$) for flight 02_STR2, either estimated from the GPS altitudes ($\delta z_{\mathrm{GPS}}$) or from the pressure observations ($\delta z_P$). Both PSDs look very similar over the whole frequency range. A large spectral peak, centered at $\sim 4 \cdot 10^{-3}$ Hz, is observed on these PSDs. This spectral peak corresponds to the SPB oscillatory motions about their EDS with period $\sim 220$ s described in Section 2.3. Apart from long-duration balloons, radar is one of the few technique that is able to infer the air vertical velocity in the free atmosphere. Few radar-borne PSDs of the vertical velocity have been published (Ecklund et al., 1986; VanZandt et al., 1991; Satheesan and Murthy, 2002). They either show a weak enhancement at frequencies close to $N$ with respect to the spectral level at $\omega \lesssim N$, or no enhancement at all. It is thus believed that the high-frequency vertical displacements in the balloon observations, 15 m in the average, mostly result from the balloon oscillating motions about their EDS, rather than from the isentropic vertical displacements of air parcels. In other words, $\delta z_B \gg \delta \zeta_\theta$ in Eqs (3) and (4).

### 3.2.2 The correlation method

The covariance of two quantities $X$ and $Y$, defined as $\mathrm{Cov}(X, Y) = \mathrm{E}[(X - \mathrm{E}[X])(Y - \mathrm{E}[Y])]$ where $\mathrm{E}$ is the mathematical expectation that is estimated by:

$$\mathrm{Cov}(\tilde{X}, Y) = \frac{1}{N-1} \sum_{i=1}^{N} (X - \bar{X})(Y - \bar{Y}) \tag{5}$$

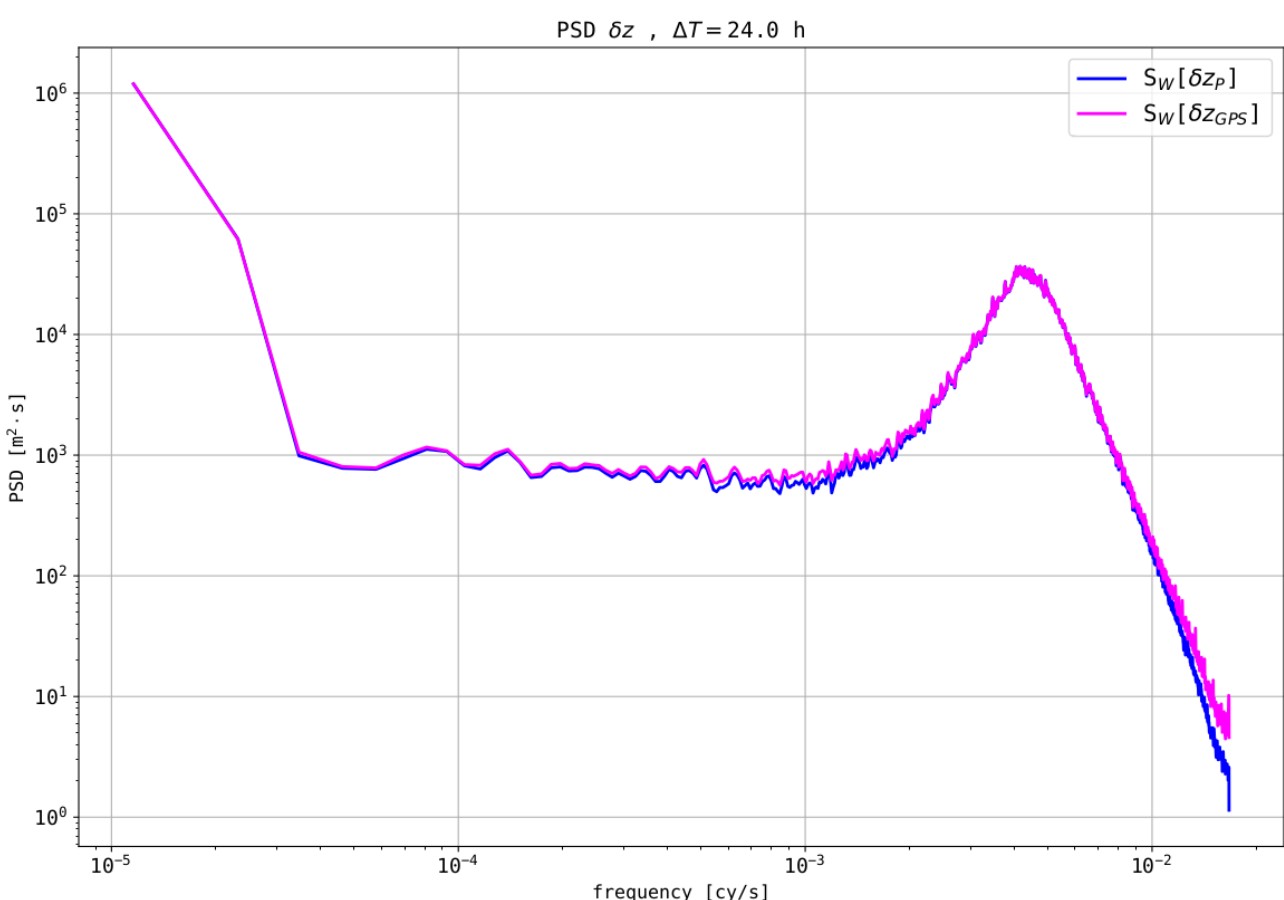

**Figure 6.** Power spectra of vertical increments of SPB heights, i.e. of vertical displacements between consecutive measurements (30s), for flight 02_STR2.

where $N$ in the sample's size. The Pearson correlation coefficient $\rho_P(X,Y) = \dfrac{\text{Cov}(X,Y)}{\sigma_X \sigma_Y}$ is estimated by:

$$r_P(X,Y) = \frac{\text{Cov}(\tilde{X},Y)}{s_X s_Y} \tag{6}$$

where $s_X$ is the estimate of the standard deviation $\sigma_X$ of quantity $X$, i.e. $s_X = \sqrt{\text{Cov}(\tilde{X},X)}$. A second correlation coefficient, the Spearman's rank correlation coefficient, $r_S(X,Y)$, has also been used in the present study. It is a measure of the statistical dependence between the rank of two variables $X$ and $Y$. It is computed as the Pearson correlation between the rank values of those two variables.

$$r_S(X,Y) = r_P(\text{rk}_X, \text{rk}_Y) \tag{7}$$

where $\text{rk}_X$ is the rank of variable $X$, i.e. the sample $X$ is is replaced by the rank of $X$ in the expression of $r_P$ (Eq. 6). The use of the non-parametric Spearman correlation makes it possible to get rid of outliers in the time series (Spearman, 1904).

If the flow is stably stratified, the increments of potential temperature $\delta\theta$ are related to the vertical displacements of the SPB $\delta z_B$ (Eq. 3). The covariance $\text{Cov}(\delta\theta, \delta z_B)$ reads:

$$\text{Cov}(\delta\theta, \delta z_B) = \text{E}[\delta\theta \times \delta z_B] = \text{E}[\bar{\theta}_z(\delta z_B \times \delta z_B - \delta z_B \times \delta\zeta_\theta)] \tag{8}$$

Noting that (i) $\delta\zeta_\theta \ll \delta z_B$, and (ii) that the vertical oscillations of the balloon and the displacements of the isentropic surfaces are expected to be non-synchronous since $\omega_B$ does not correspond to any atmospheric frequency ($\omega_B > N$), we assume that the covariance $\text{E}[\delta z_B \times \delta\zeta_\theta]$ is negligible compared to the variance of $\delta z_B$, i.e.

$$\text{Cov}(\delta\theta, \delta z_B) = \text{E}[\bar{\theta}_z \delta z_B \times \delta z_B] \tag{9}$$

Notice that under the above hypothesis, the covariance of a non-conservative measured quantity $X$ ($T$, $u$, $v$) reads:

$$\text{Cov}(\delta X, \delta z_B) = \text{E}[\bar{X}_z \delta z_B \times \delta z_B]. \tag{10}$$

If $\bar{\theta}_z$ is positive in the time interval during which the covariance is estimated, $\text{Cov}(\delta\theta, \delta z_B)$ is expected to be positive. Moreover, if $\bar{\theta}_z$ is strictly constant during the time interval:

$$\bar{\theta}_z = \frac{\text{Cov}(\delta\theta, \delta z_B)}{\text{Var}[\delta z_B]} \tag{11}$$

If $\bar{\theta}_z$ is not strictly constant, as it is very likely the case, the ratio $\text{Cov}(\delta\theta, \delta z_B)/\text{Var}[\delta z_B]$ can be interpreted as an estimate of the mean gradient $\langle \bar{\theta}_z \rangle$ during the considered time interval at the flight level of the SPB. For a non-conservative quantities $X$:

$$\bar{X}_z = \frac{\text{Cov}(\delta X, \delta z_B)}{\text{Var}[\delta z_B]} \tag{12}$$

In the ideal case where $\bar{\theta}_z$ constant and there is no instrumental noise, $\sigma_{\delta\theta} = \bar{\theta}_z \sigma_{\delta z_B}$ and the correlation coefficient $r_P(\delta\theta, \delta z_B) = 1$, whatever $\bar{\theta}_z > 0$. This conclusion also holds for the Spearman correlation $r_S$ if a linear relation is assumed between $\delta\theta$ and

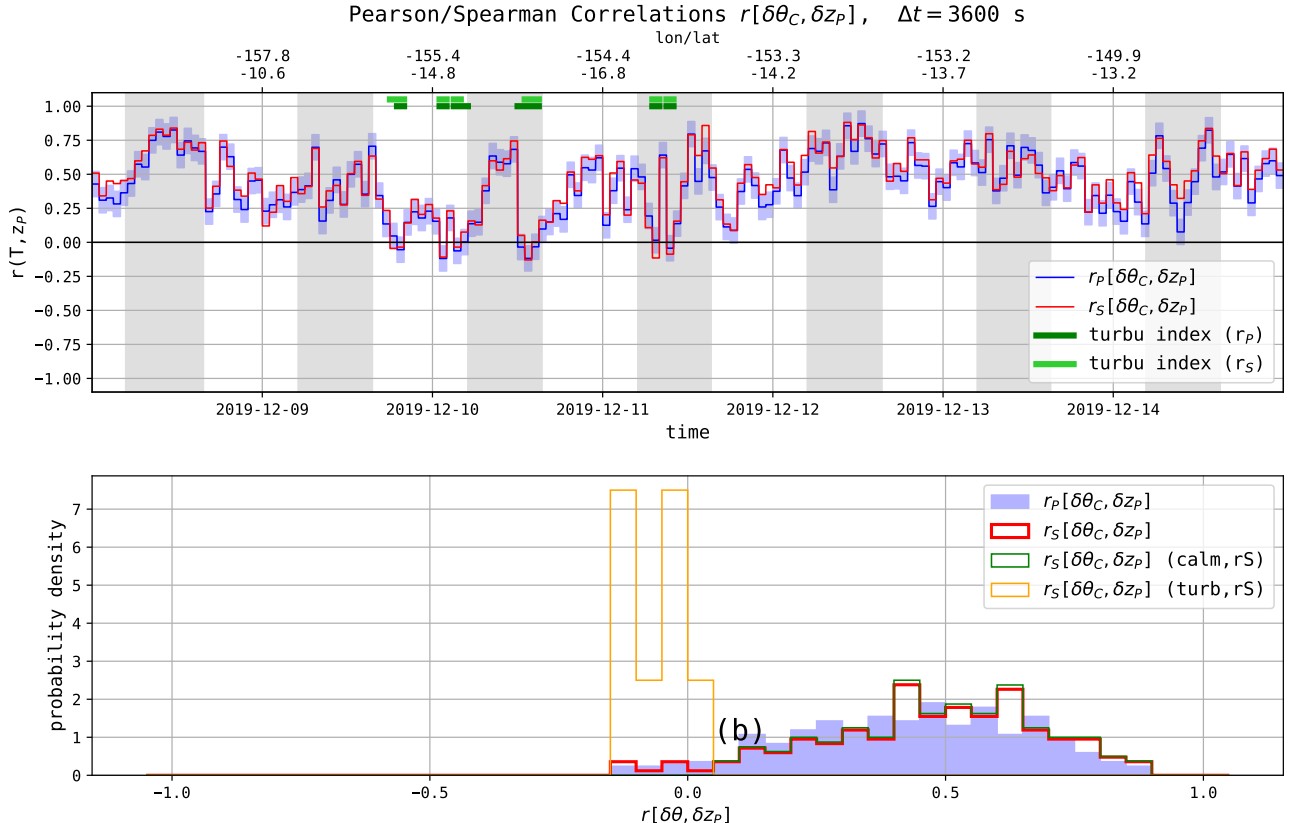

**Figure 7.** Top: time series of correlation coefficients $r_P$ and $r_S$ applied to the increments potential temperature derived from $T_C$ and vertical displacements from $z_P$. The correlations are evaluated on one-hour time segments (120 samples). The time series runs for seven days, from 08/12/2019 to 15/12/2019, the grey vertical stripes indicating nighttime. The light-blue shaded areas show the $\pm$ one standard deviation interval for the Pearson correlations. Bottom: histograms of the two correlation coefficients obtained during the week. The blue (filled) and red (thick line) correspond to all the one-hour time segments, the thin green and orange lines correspond to the calm (non-turbulent) and turbulent time intervals.

$\delta z_B$. if $\bar{\theta}_z > 0$ but is no more constant during the considered time interval, the correlation coefficient is smaller than one but is still positive since the product $\delta\theta \times \delta z_B$ is always positive. The correlation is further reduced, but remains positive, in the presence of uncorrelated noise (see appendix).

The first method to detect turbulent mixing is to check for a non-positive correlation, i.e. a null or negative correlation, between $\delta\theta$ and $\delta z_B$. In order to infer if the null hypothesis $H_0$ has to be rejected, that is if a positive correlation exists, a standard hypothesis test is performed. A confidence interval for the correlation coefficient is based on a Fisher transformation of the correlation estimate (Hotelling, 1953). The $H_0$ hypothesis is here rejected if the correlation estimate exceeds half of its standard deviation (unilateral test), the corresponding confidence level being 69.15%. If the $H_0$ hypothesis cannot be rejected, turbulent mixing is diagnosed, and the turbulence flag is set to 'True'. Two turbulence indices are built, either from the Pearson or Spearman correlations

Figure (7) illustrates the time series (top panel) and histograms (bottom panel) of $r_P$ and $r_S$ during one week of flight 02_STR2. Both correlations are estimated over time segments of one hour. The light-blue shaded areas show the $\pm$ one standard deviation interval for the Pearson correlations only. The thick green lines at the top of the upper panel display the time interval during which turbulence is detected, i.e. when the turbulence indices are set to true.

As expected, the correlation coefficients are found to be positive most of the time. On a few occurrences during the time period shown in Figure (7), turbulence is detected as the correlation coefficients approach zero. A good agreement in the time periods identified as turbulent is found whatever the correlation method used.

### 3.2.3 The impact of instrumental noise on the correlation coefficients

The impact of measurement noise can be evaluated through the averages of the largest correlation coefficients, all corresponding a priori to $\bar{\theta}_z > 0$. Recall actually that, without instrumental noise, the correlation $(\delta\theta, \delta z_B)$ is ideally 1 if $\bar{\theta}_z > 0$ and keeps constant over 1 hr. As shown in Equation (B1) of the Appendix, it is the ratio of the instrumental noise to the variance of the "geophysical" signal that appears in the correlation coefficient. In other words, the larger the amplitude of the balloon oscillations about their EDS, the less the measurement noise reduces the correlation coefficient.

The Spearman and Pearson correlation coefficients have been estimated for both potential temperatures increments, $\delta\theta_C$ and $\delta\theta_S$, and for both altitudes increments, $\delta z_P = -\delta P/\rho_C$ and $\delta z_{\text{GPS}}$. Four turbulence indexes coefficients are thus estimated. We have then averaged the 50% largest correlation coefficients, i.e. the quantiles 50-100 of the coefficients, for each of the eight flights (table 4). These averaged coefficients give information, at least relatively, on the quality of the data and on the performance of the turbulence estimators: the larger the correlations, the smaller the impact of instrumental noise.

Table 4 again illustrates that Flight 03_TTL3 is abnormally noisy because of the poorer (potential) temperature measurements (cf Table 2). Systematically, we observe that:

1. the correlations are larger when using $\theta_C$ rather than $\theta_S$,

2. there is a slight amelioration in using $\delta z_P$ rather than $\delta z_{\text{GPS}}$,

3. the Spearman correlations ($r_S$) are slightly larger than the Pearson correlations ($r_P$).

We shall therefore prefer the Spearman estimator, using $\delta T_C$ (and derived $\delta\theta_C$) combined with $\delta z_P$ in evaluating both the correlations and the vertical gradients from which the turbulent indices are estimated, as the impact of instrumental noise appears to be less in this estimator.

| Flight Id | $r_S$ $\delta\theta_S, \delta z_P$ | $r_P$ $\delta\theta_S, \delta z_P$ | $r_S$ $\delta\theta_S, \delta z_{\text{GPS}}$ | $r_P$ $\delta\theta_S, \delta z_{\text{GPS}}$ | $r_S$ $\delta\theta_C, \delta z_P$ | $r_P$ $\delta\theta_C, \delta z_P$ | $r_S$ $\delta\theta_C, \delta z_{\text{GPS}}$ | $r_P$ $\delta\theta_C, \delta z_{\text{GPS}}$ |
|---|---|---|---|---|---|---|---|---|
| 01_STR1 | 0.563 | 0.566 | 0.544 | 0.540 | **0.592** | 0.589 | 0.570 | 0.561 |
| 02_STR2 | 0.640 | 0.638 | 0.643 | 0.640 | 0.681 | 0.671 | **0.683** | 0.673 |
| 03_TTL3 | 0.402 | 0.284 | 0.392 | 0.278 | **0.464** | 0.319 | 0.457 | 0.315 |
| 04_TTL1 | 0.614 | 0.604 | 0.603 | 0.588 | **0.678** | 0.646 | 0.665 | 0.630 |
| 05_TTL2 | 0.605 | 0.572 | 0.600 | 0.566 | **0.668** | 0.657 | 0.661 | 0.648 |
| 06_STR1 | 0.569 | 0.549 | 0.547 | 0.524 | **0.622** | 0.593 | 0.603 | 0.569 |
| 07_STR2 | 0.585 | 0.566 | 0.563 | 0.537 | **0.587** | 0.556 | 0.564 | 0.527 |
| 08_STR2 | 0.639 | 0.631 | 0.638 | 0.629 | **0.699** | 0.685 | 0.697 | 0.683 |

**Table 4.** Averages of the 50% largest correlation coefficients between $\delta\theta$ and $\delta z$. Spearman and Pearson coefficients are displayed for the four combinations between $\delta\theta_S$, $\delta\theta_C$ and $\delta z_P, \delta z_{\text{GPS}}$. About 1200 coefficients are here averaged for each flight. The largest correlation coefficient is displayed in bold for each flight. The larger the correlations, the smaller the impact of instrumental noise (see text).

### 3.2.4 The Richardson method

The second detection method is based on an evaluation of the local Richardson number $Ri$. Notice that $\bar{\theta}_z$ (Eq. 11) is nothing else that the slope of a least square linear regression between the 120 potential-temperature ($\delta\theta$) and vertical-displacement increments ($\delta z_B$). This method of estimating the vertical gradient of the potential temperature is labeled the least square fit (LSF) method in the following. An alternative fitting method, the so-called Theil-Sen fitting (TSF) method has also been used (Sen, 1968). The Theil-Sen estimator is defined as the median of the slopes of all lines through pair of points of the sample. It is a non-parametric and robust method, almost insensitive to outliers. Both fitting methods have been applied to estimate the vertical gradients of several measured quantities, such as $T$, $P$ or horizontal wind velocities $u$ and $v$ (Eq.12).

Two estimates of the mean Brunt-Väisälä frequency are thus computed with either the LSF or the TSF, applied to $\delta T_C$ and $\delta z_P$, which were respectively identified as the less noisy estimates of the air temperature and vertical displacements (see Tables 2 and 3). The squared shear, evaluated from the vertical gradients of the horizontal velocities, is obtained similarly. Two turbulence indices are hence built by using LSF and TSF estimates of the Richardson number, and are set to 'True' if $Ri \leq 1/4$.

The upper panel of Figure (8) displays the vertical gradients of temperature for the same one-week time series than in Figure 7. Estimates are shown for only the thermocouple temperatures and vertical displacements obtained with the pressure increments. Time series (not shown) obtained from other pairs of variables (by using $T_S$ or $T_C$, $z_{\text{GPS}}$ or $z_P$) are very similar to the one presented, whatever the used fitting method. The uncertainty ($\pm$ one standard deviation) is shown in light-red for the LSF estimator, as well as the turbulence detection criteria based on the Richardson number (thick green lines at the top of the figure). The bottom panel of Figure (8) shows the probability distribution functions (PDF) of the vertical gradients of temperature shown above (filled in blue and thick curve). PDFs corresponding to turbulent and calm (i.e. non-turbulent) periods according to the Richardson number criterion are also displayed in thin lines.

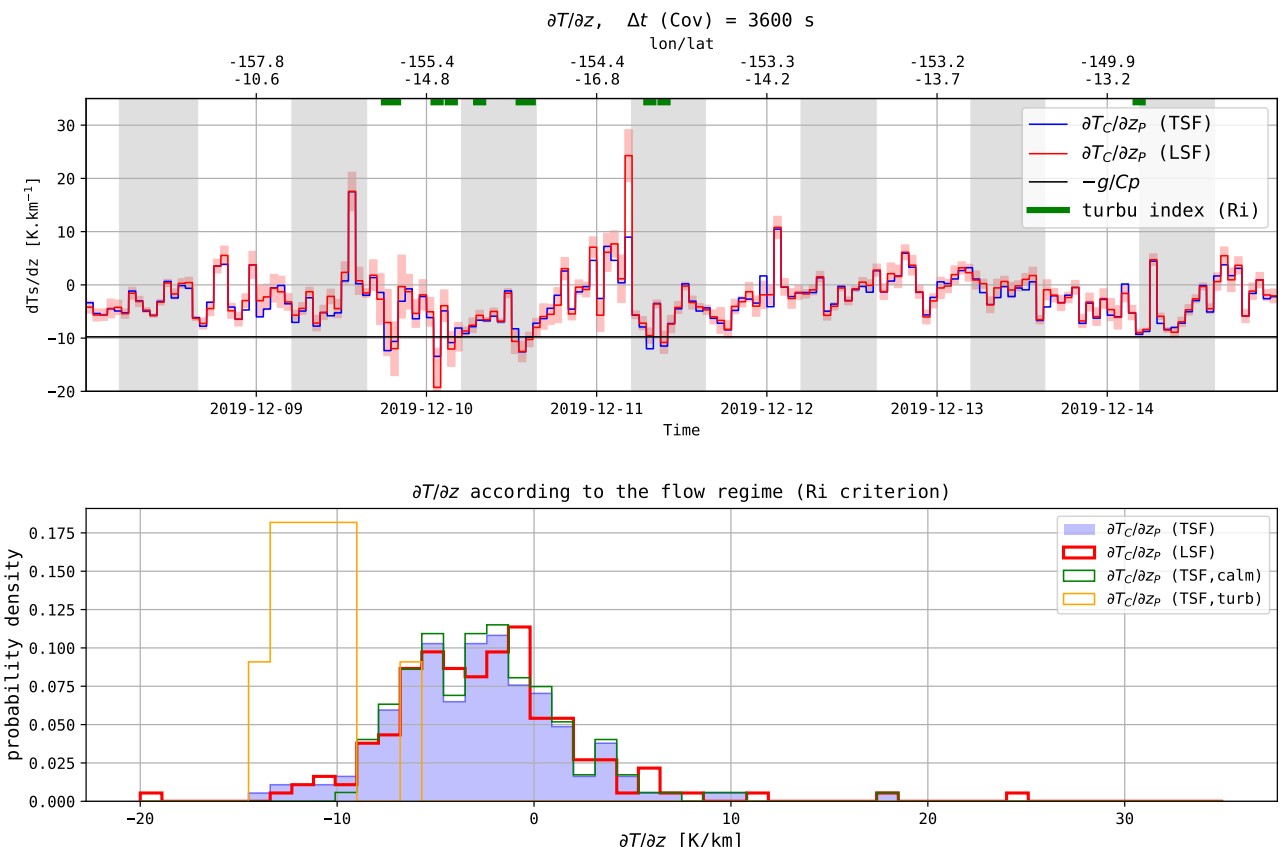

**Figure 8.** Top: time series of $\partial T/\partial z$ estimated by two fitting methods, least square (LSF) and Theil-Sen (TSF) from $T_C$ and $z_P$. Linear fits are performed on one-hour time segments (120 samples). The light-red shaded areas show the $\pm$ one standard deviation interval for the LSF estimates. Bottom: histograms of the temperature gradients by the two methods. The histograms corresponding to the turbulent and laminar time segments are also plotted (light curves).

First, it should be noticed that the choice of the LSF or TSF estimates for the vertical gradient of temperature or for the turbulence index has only a minor impact. Similarly, the choice of parameters ($z_P$ or $z_{\mathrm{GPS}}$, $T_C$ or $T_S$) and any combination of those has little influence on the detection of turbulence layers (not shown). Figure 8 also clearly illustrates that the time periods flagged as turbulent by the Richardson number index are associated with the small-value tail of the temperature-gradient PDF ($dT/dz \lesssim -10$ K/km). Last, those time periods are essentially the same than the one shown in Figure 7, which have been

obtained with the correlation method.

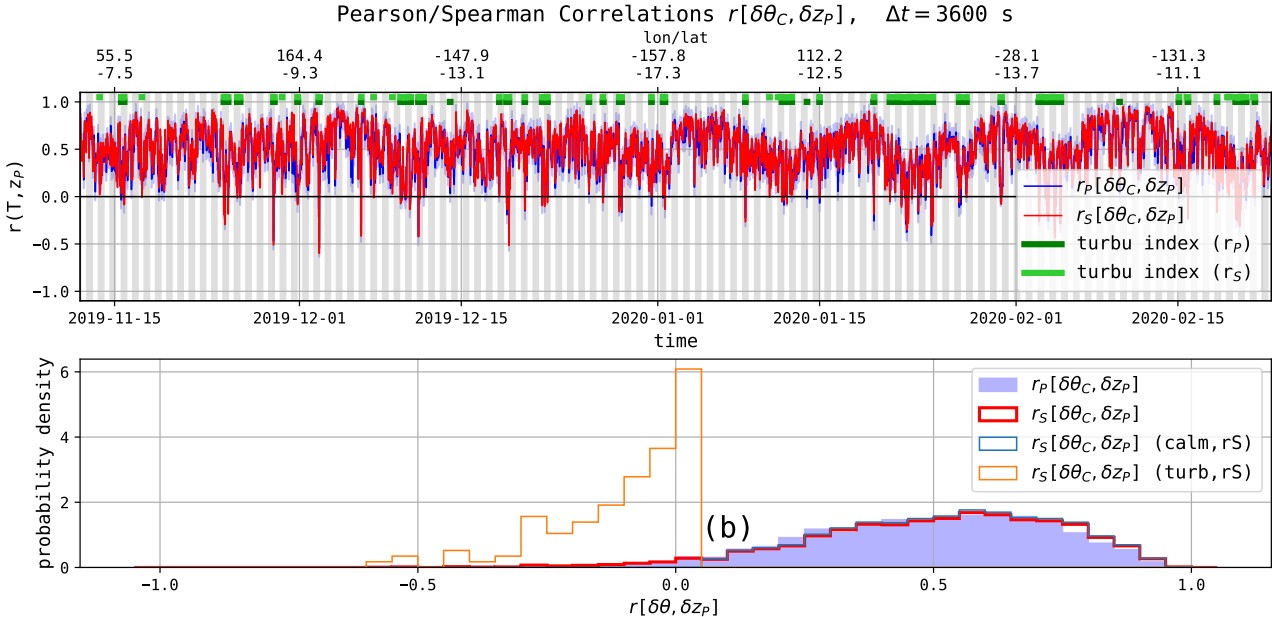

**Figure 9.** Top: time series of the Spearman and Pearson correlation coefficients between $\delta\theta_C$ and $\delta z_P$ during flight 02_STR2 (103 days). The turbulence indexes deduced from the two correlations are shown (thick green lines at the top of the plot). Bottom: Histograms of the Spearman (thick line) and Pearson (filled) coefficients for flight 02_STR2. Also shown, the histograms corresponding to the turbulent and laminar cases (thin lines). The time series and histograms of the two correlation coefficients are very similar even though few differences in turbulence detection are visible.

## 4 Results

### 4.1 Turbulence detection with the correlation method

The top panel of figure 9 displays the time series of both the Spearman and Pearson correlations coefficients between $\delta\theta_C$ and $\delta z_P$ for the whole flight 02_STR2. As previously, the correlations are estimated on time segments of one hour (120 samples).

They are ranging from $\sim -0.5$ to $0.96$. Both time series exhibit very similar variations, showing time intervals lasting several days with relatively large correlations ($> 0.7$) and short bursts of time with low or even negative correlations. Two turbulence indices inferred from the Spearman and Pearson correlation methods are shown as thick green lines at the top of the plot. The fraction of time during which the flow is detected turbulent for flight 02_STR2 is 4.5% (resp. 4.4%) from the Spearman (resp. Pearson) correlations (see Table 5). Time series and histograms of correlations obtained by using measurement of $\delta\theta_S$ or $\delta z_{\mathrm{GPS}}$

(not shown) provide time series that are almost indistinguishable from those shown in Figure 9.

The bottom panel of figure 9 shows the PDFs of the two correlation coefficients over the whole flight (filled in blue and thick red line). Both distributions are observed to be almost identical, although $r_S$ is slightly, and systematically, larger than $r_P$ for correlations larger than $0.5$. The average values for both correlation coefficients are close: $\overline{r_S[\delta\theta_C, \delta z_P]} = 0.50$ and $\overline{r_P[\delta\theta_C, \delta z_P]} = 0.48$. The thin lines show the PDFs of $r_S$ for turbulent-only (orange) and laminar-only time periods (green).

The PDFs associated with the laminar time periods are almost identical to those of the whole flight, since they correspond to about 95% of the data. The turbulent time periods correspond only to the small-value tail of the overall distribution.

The fractions of time during which the flow is found turbulent with the correlation method are reported in Table 5 for the eight flights of the campaign. They look overall very consistent, typically ranging between $4.5\%$ and $6.5\%$, with the exception of flight 03_TTL3, which is associated with noisier temperature measurements.

| Flight Id | $r_S(\delta\theta_C, \delta z_P)$ | $r_P(\delta\theta_C, \delta z_P)$ | $Ri(\delta\theta_C, \delta z_P)$ |
|---|---|---|---|
| 01_STR1 | 5.1 | 4.9 | 3.7 |
| 02_STR2 | 4.5 | 4.4 | 3.7 |
| 03_TTL3 | 11.2 | 21.5 | 15.6 |
| 04_TTL1 | 3.8 | 4.9 | 3.4 |
| 05_TTL2 | 6.6 | 6.6 | 5.7 |
| 06_STR1 | 5.8 | 6.6 | 4.3 |
| 07_STR2 | 5.5 | 6.3 | 3.3 |
| 08_STR2 | 5.5 | 6.2 | 5.7 |

**Table 5.** Fraction of time (percent) during which the flow is found to be turbulent. Turbulence is diagnosed from the correlation method based of $\delta\theta_C$ and $\delta z_P$ (two first columns) and from the $Ri(\delta T_C, \delta z_P)$ criterion (last column).

## 4.2 Estimations of the vertical temperature gradient with fitting methods

Figure 10 shows estimates of $\partial T/\partial z$ for flight 02_STR2 inferred from the linear fitting of the increments of temperature and vertical displacements during hourly intervals. More precisely, the upper panel of this figure shows the vertical temperature gradient obtained with the TSF method applied to $\delta T_C$ and $\delta z_P$. Time series obtained by using other combinations of measured variables and fitting methods (not shown) are almost indistinguishable from the one shown in Figure 10. The dry adiabatic lapse rate $\Gamma = -g/c_P$ is also indicated as a black line. The turbulence index based on $Ri$ is shown as a thick green discontinuous line at the top of the upper panel. The lower left panel shows the PDFs of $\partial T/\partial z$ for the whole flight, and by distinguishing calm and turbulent time periods. The distribution of $\partial T/\partial z$ is found very asymmetric, the mode being close to $-4$K/km. Most of the turbulent cases are associated with $\partial T/\partial z < \Gamma$, few of them being associated with $\partial T/\partial z > \Gamma$. The cumulative distribution function (CDF) of the temperature gradients are displayed on the lower-left panel. About 80% of the detected turbulent cases, associated with $Ri < 0.25$, correspond to super-adiabatic temperature gradient.

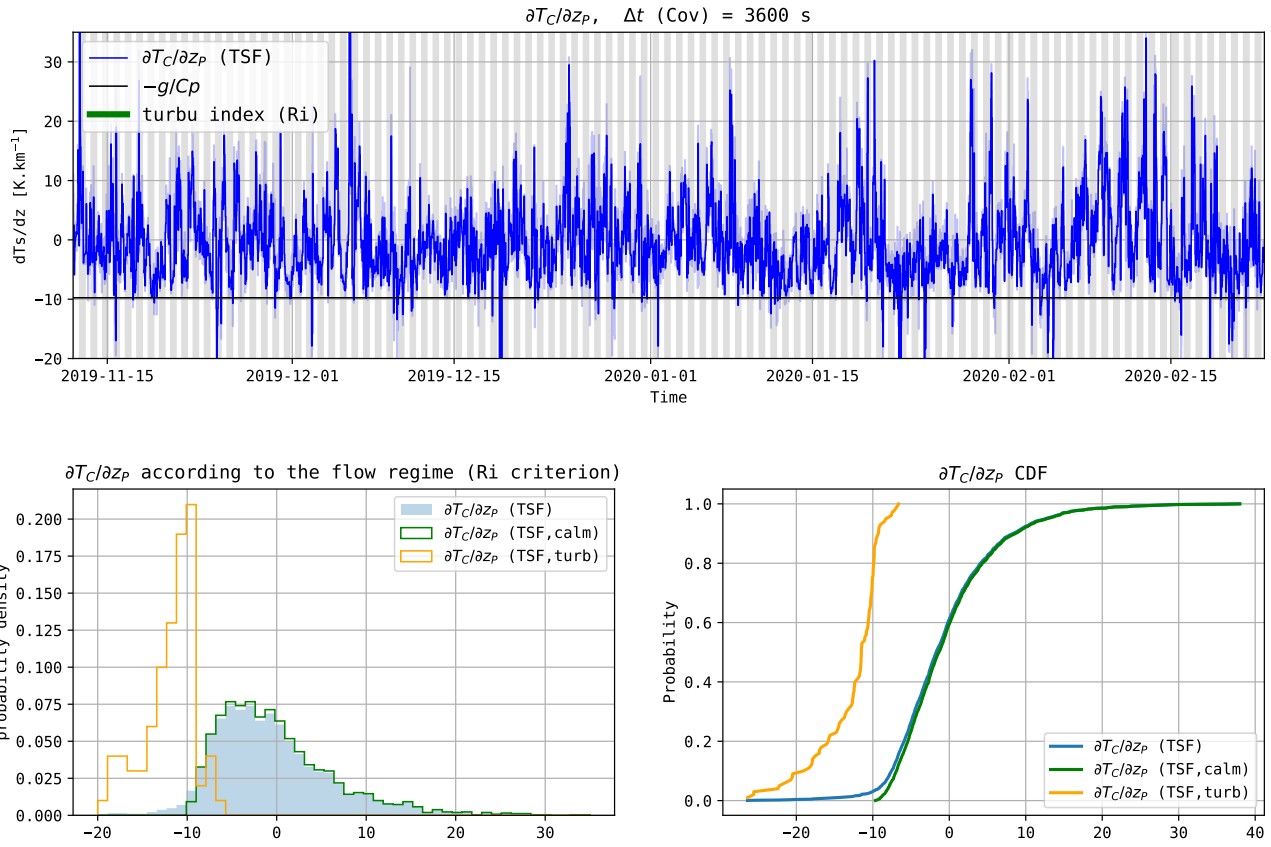

**Figure 10.** Top panel: two estimates of $\partial T_C/\partial z_P$ for flight 02_STR2, from the linear fitting of $T_C$ and $\delta z_P$ on hourly intervals. The black line shows the adiabatic lapse rate $-g/c_P$. The $Ri$ turbulence index is drawn on the top of the figure as a thick green discontinuous line. Bottom left: Histograms of $\partial T_C/\partial z_P$ for all the time intervals (filled in blue) and by distinguishing the laminar and turbulent cases (thin lines). Bottom right: CDF of the temperature gradients.

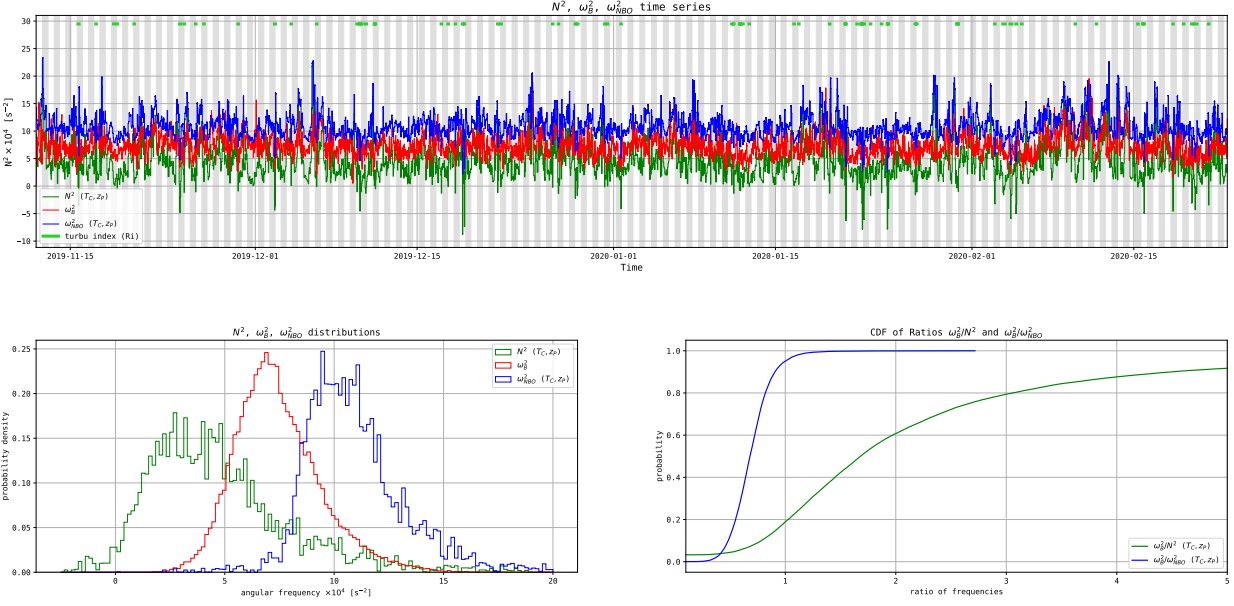

**Figure 11.** Brunt-Väisälä, NBO and observed balloon frequencies. Brunt-Väisälä and NBO frequencies are deduced from the estimates of $\partial T/\partial z$ from $\mathrm{Cov}(\delta T_C, \delta z_P)$: times series (top), histograms (lower left), CDF of the ratios $\omega_B^2/N^2$ and $\omega_B^2/\omega_{\mathrm{NBO}}^2$ (lower right).

Based on $\partial T/\partial z$ estimates, the Brunt-Väisälä and NBO frequencies can be evaluated, cf. Eqs. (1) and (2). Figure 11 displays the squared Brunt-Väisälä ($N^2$), theoretical NBO ($\omega_{\mathrm{NBO}}^2$) and observed balloon ($\omega_B^2$) frequencies for the whole 02_STR2 flight.
Brunt-Väisälä and NBO frequencies are deduced from the hourly estimates of $\partial T/\partial z$. The observed balloon frequencies are directly estimated from the observations of the balloon oscillations (Fig. 3). A 60 min running average is then applied to the raw $\omega_B$ time series. The three frequencies are close, but distinct, to each other, the observed balloon frequency ranging between $N$ and $\omega_{\mathrm{NBO}}$. The CDF of frequency ratios (lower right panel) reveal that $\sim 95\%$ of the measured $\omega_B$ are smaller than the $\omega_{\mathrm{NBO}}$ estimates, and that 19% of $\omega_B$ are smaller than $N$. This property is consistent with numerical simulations of the motion of a
spherical SPB assuming atmospheric forcing occurs at frequencies lower than the Brunt-Väisälä frequency (Podglajen et al., 2016, in particular see the supplementary information).

Figure (12) shows the histograms of $T_z = \partial T_C/\partial z_P$ for flight 3 obtained from the TSEN measurements (TSF method) and from the RACHuTS temperature profiles - down to two kilometers below the balloon. The vertical gradients of the RACHuTS temperature are estimated on 30 m vertical segments. The RACHuTS histogram has common features with the TSEN his-
395 tograms: asymmetric distribution, same negative modes, sharp transition around -10$^o$C/km. These distributions are considered to be consistent despite the fact that they are not obtained in the same altitude domain and not simultaneously.

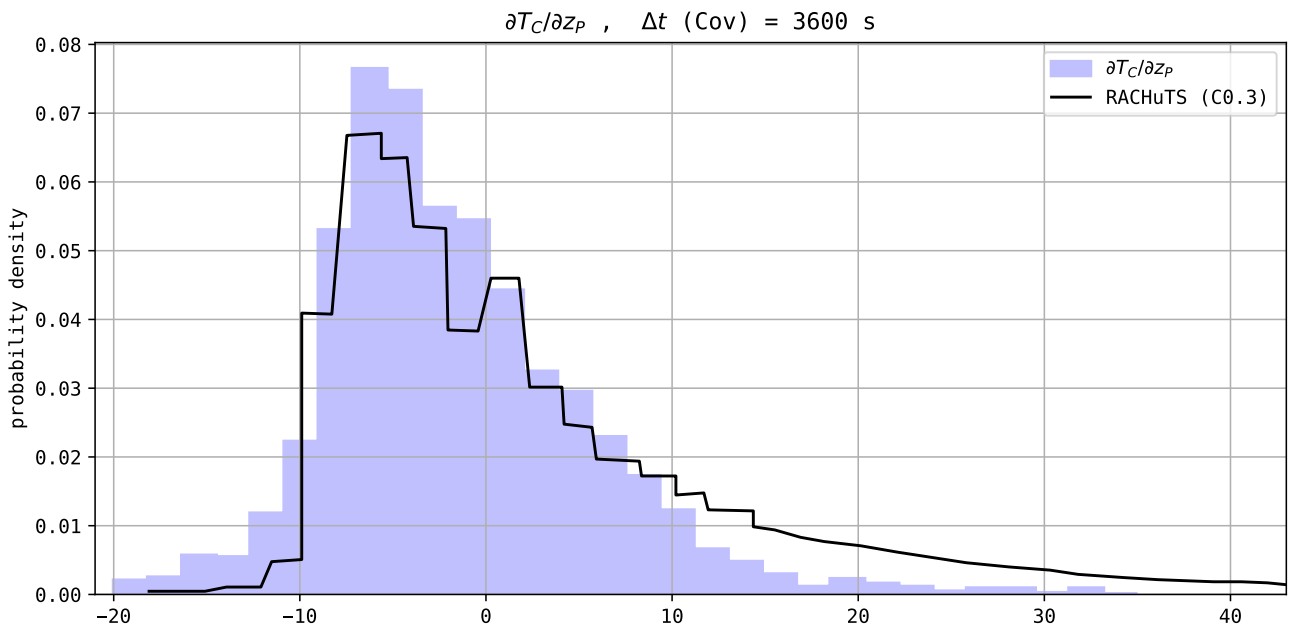

**Figure 12.** Histograms of $\partial T_C/\partial z_P$ from the TSEN measurements (flight 03_TTL3) and from the RACHuTS temperature profiles are superimposed. The two histograms have common characteristics despite the fact that they are not obtained at the same altitude levels.

### 4.3 Turbulence detection with the $Ri$ method

Figure 13 shows estimates of the Richardson number along Flight 02_STR2 obtained either by the LSF or TSF estimates with $T_C$, $z_P$, $u$ and $v$ observations. The corresponding two turbulence indices ($Ri \leqslant 1/4$) are also shown on the top of the upper panel (green thick line). The bottom plot of Figure 13 displays the histograms of the two $Ri$ estimates. The time series and histograms of the two Ri estimates look very similar even though few differences in the turbulence detection are visible.

As shown in Table 5, the fraction of time during which the flow is found turbulent according to the $Ri$ criterion is 3.7% on Flight 02_STR2. More generally, the detection of turbulent flow during the Strateole-2 C0 campaign ranges from 3.3 to 5.7% of the time, with the exception of Flight 03_TTL3.

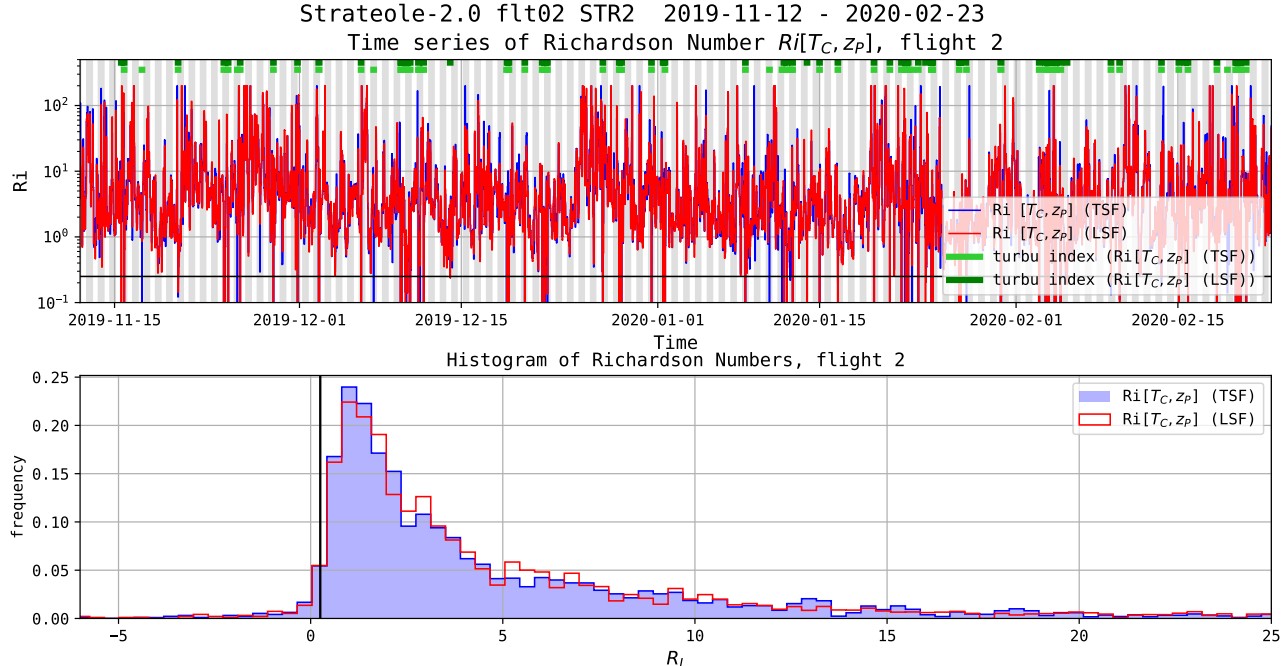

**Figure 13.** Top panel: Richardson number estimates from $T_C$, $u$, $v$ and $z_P$, by using the least square fitting (LSF) and Theil-Sen fitting (TSF) methods. The thick black line shows the threshold Ri = 0.25. The two inferred turbulence indexes are shown (thick green lines at the top of the plot). Bottom panel: histograms of the two Richardson numbers estimates.

## 5 Discussion

### 5.1 Possible impact of wake on the temperature measurements

Due to the vertical oscillations of the balloon, the T-sensors are possibly in the wakes of the balloon or of the chain flight. Note that we expect the balloon wake to be warm during daytime and cold during nighttime, the balloon being cooler than the ambient air during nighttime.

The temperature sensors are located 27 m below the balloon base (except for 03_TTL3 flight) and 15 m below the EUROS gondola. The diameter of the balloons is either 11 m (TTL) or 13 m (STR). On all but TTL3 flights, the T-sensors are located 7 m below the last gondola in the flight chain. Flight 03_TTL3, carrying the RACHuTS system, is an exception since the temperature sensors are located 30 cm away from the EUROS gondola.

The question of the possible impact of the wake on temperature measurements has been taken into account. Indeed, we have calculated the statistics, vertical gradients and correlations, considering only the downward phases of the oscillations, i.e. when the temperature sensors (which are located at the lower end of the flight chain) sample the "fresh" air if a minimum shear exists. The resulting time series (not shown) are noisier since we only consider about half of the samples. However, both time

|  | $r_P[T_C,z_P]$ | $r_S[T_C,z_P]$ | $Ri_{\text{LSF}}[T_C,z_P]$ | $Ri_{\text{TSF}}[T_C,z_P]$ |
|---|---|---|---|---|
| $r_P[T_C,z_P]$ | 100 | 98.95 | 98.46 | 98.42 |
| $r_S[T_C,z_P]$ |  | 100 | 97.90 | 98.59 |
| $Ri_{\text{LSF}}[T_C,z_P]$ |  |  | 100 | 99.07 |

**Table 6.** Percentage of identical turbulence detections from four turbulence index by using $(Tc, zP)$ for flight 02_STR2

series of correlations and temperature gradients have similar characteristics to those calculated when considering all samples, showing the same succession of stable and unstable periods. We therefore conclude that the impact of the wake does not affect

significantly the estimated statistics, correlations and covariances, and because of the increase of noise we choose to consider all the samples.

## 5.2   Comparison between the turbulence indexes

Table 6 shows the percentage of identical detections, laminar or turbulent, of the four turbulence indexes, $r_P$, $r_S$, $Ri_{\text{LSF}}$ and $Ri_{\text{TSF}}$, for the eight balloon flights. The percentage of similar detections ranges from 97.9% ($r_S$ vs. $Ri_{\text{LSF}}$) to 99.07% . How-

ever, as pointed out by an anonymous reviewer, such overall agreement does not imply equally good agreement between estimators when turbulence is detected. Choosing $Ri_{\text{TSF}}$ as a reference, we compared the diagnoses with the other three estimators, $Ri_{\text{LSF}}$, $r_P$ and $r_S$. Figure 14 shows the percentages of similar and different detections when the flow is diagnosed as turbulent (T) or laminar (L). It reveals that the detections are similar for more than 99% of the cases if the flow is diagnosed as laminar. If the flow is diagnosed as turbulent, the rate of identical detections drops to about 80%, the comparison being the

worst for $r_P$ (76%), and comparable for $r_S$ and $Ri_{\text{LSF}}$ (respectively 83.2% and 85.4%). We believe that these differences result mainly from the fact that the threshold values, zero correlation or Ri = 0.25, correspond to the tails of the distributions of these estimates (see the histograms of Figs 9 and 13). When the atmosphere is weakly stratified, threshold effects are likely to be important, leading to some differences in the diagnoses of flow conditions.

We also found that the time fraction of turbulent episodes obtained by the $Ri$ criterion is almost always smaller than that

obtained by the correlation methods (Table 5). This can be partly due to the thresholds values of the hypothesis tests of a null correlation (i.e. to the choice of a confidence interval).

## 5.3   Occurrences of negative values for $Ri$ and $N^2$

The time series of $N^2$ (Fig. 11) and $Ri$ (Fig. 13) show some negative values. For the considered flight (02_STR2), the occurrence frequency of negative $N^2$ is 3.4% (from the Theil-Sen regression performed on $T_C$ and $z_P$). Such negative $N^2$ (of

$Ri$) can result from both the dispersion of the temperature gradients estimates and the occurrences of episodes of unstable stratification.

Negative estimates of $N^2$ ($Ri$) could be due to the precision of the temperature gradients estimates which are expected to be scattered around a value close to $-10^o$/km in case of neutral stratification. Temperature gradients are estimated from the

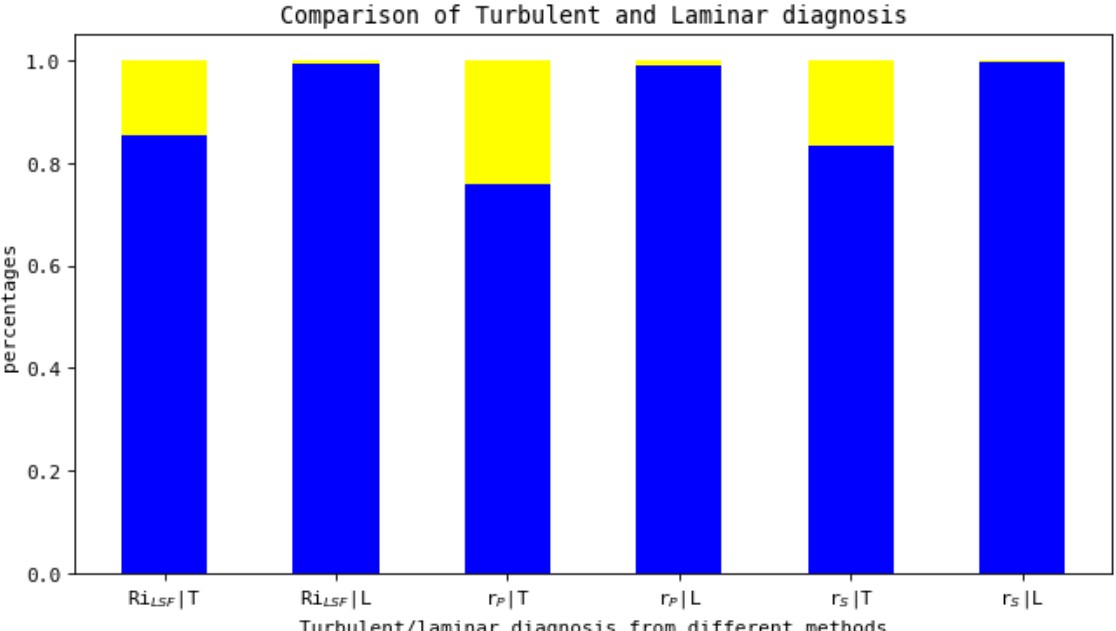

**Figure 14.** Percentages of true (similar) and false (different) detections for turbulent (T) and laminar (L) episodes diagnosed by the $Ri_{\mathrm{TSF}}$ criterion. The three estimators $Ri_{\mathrm{LSF}}$, $r_P$ and $r_S$ are compared to $Ri_{\mathrm{TSF}}$. The rate of agreement is larger than 99% in case of laminar flow, about 80% in case of turbulent flow.

covariance of temperature increments and displacements, with these covariances scattered around their mean values. As a result
$N^2$ estimates can be negative even if the stratification is neutral, or nearly neutral.

However, unstable stratifications ($N^2 < 0$) seem to occur in the lower stratosphere since they have been reported in the literature. For instance, detection of turbulence by the Thorpe method from in-situ measurements is based on observations of $\partial\theta/\partial z < 0$, i.e. $N^2 < 0$ (Thorpe, 1977). The probability of occurrence of such unstable layers likely depends on the vertical resolution of the profiles (see for instance Wilson et al., 2011) but it not zero. In the lower stratosphère, KHI are expected to
be the main source of instability. For KHI, turbulence is expected to be triggered for $0 < Ri < 1/4$, i.e. for $N^2 > 0$, but once it is developed, the stratification can become almost neutral ($N^2 \approx 0$), or even unstable ($N^2 < 0$), as a result of stirring and mixing. Therefore, it is plausible that the occurrences of unstable episodes may also contribute to negative values for $N^2$, or $Ri$, estimated from covariances.

### 5.4   Spatial inhomogeneity of turbulence detections

The detection of turbulent episodes is far from uniform over the globe. Figure 15 shows the positions of the turbulence detections as green dots for the eight flights of the C0 campaign. Turbulence detections seem to be very rare over some regions,

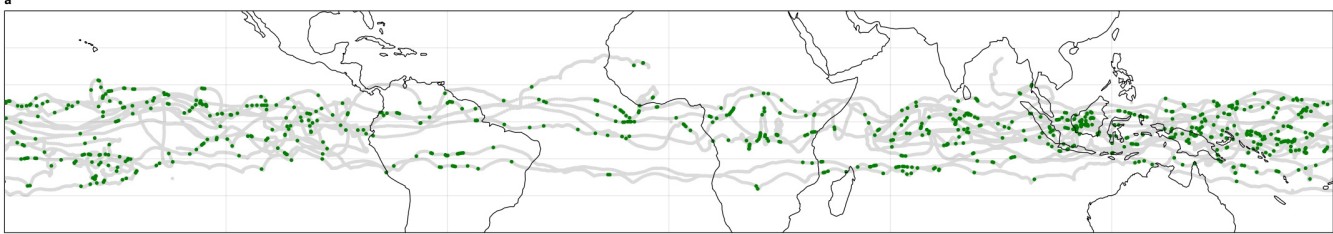

**Figure 15.** The positions of the turbulent patches are shown as green dots for the eight flights of the C0 campaign. Each dot corresponds to a one hour time interval. The detections are based on the $Ri_{\mathrm{TSF}}$ criterion. The detected turbulence episodes are far from uniform around the globe.

e.g. the South Atlantic Ocean, and quite frequent over other regions, e.g. the maritime continent or the western Pacific. At the present stage, this result is only qualitative, but interestingly these inhomogeneities are likely related to the causes of the turbulence occurrences. The study of the processes at the origin of the turbulence episodes in the UTLS is in progress.

## 6    Summary and concluding remarks

The present paper dealt with the detection of turbulence on superpressure balloon flights that drift for several months in the UTLS. During the Strateole-2 C0 campaign, eight SPBs were launched. Statistical methods to infer the flow regime, either laminar or turbulent, in which the SPB is drifting are described. Some properties of the local stratification of the flow are also inferred. These methods are based on the in-situ GPS altitude, pressure and temperature measurements, which are performed on all the SPBs of the Strateole-2 C0 campaign with a time resolution of 30 s.

We make use of the SPB oscillations about their equilibrium density surface, which enable SPBs to vertically explore the atmosphere over typically 30 m (peak-to-peak displacements). The observed periods of oscillation, $\sim 220$ s in the average, are significantly smaller than the Brunt-Väisälä period. The large amplitude of these balloon motions at frequencies higher than the Brunt-Väisälä frequency (where turbulence is expected to occur) makes it very difficult if not impossible, to detect turbulence from the direct characterization (i.e. from the variance or power spectra) of high-frequency fluctuations. On an other hand, thanks to the vertical motions of the SPBs around their EDS, the vertical gradients of any measured quantity can be estimated, either through the covariance between the increments of this quantity and those of the vertical displacements, or equivalently through the linear fit between these two increments. For the present study, covariances and linear fits have been estimated on data segments of 1 hr, i.e. 120 data samples.

Several turbulence indices (true or false) are defined and compared. A first index is based on an inference test on the correlation between potential temperature and altitude increments. A null correlation is expected in the case of turbulent mixing, since the vertical gradient of potential temperature tends to zero. Two correlation coefficients, the Pearson and Spearman

coefficients, have been tested and compared for the eight balloon flights. Alternatively, based on a linear fit between increments, vertical temperature gradients and horizontal wind can be evaluated on one-hour intervals, from which the local Richardson

number can be deduced. A second turbulence index is then based on the criterion $Ri < 0.25$. These different turbulence indices compare well since they coincide for more than 97% of the cases.

SPBs sample the atmosphere without any spatial or temporal sampling bias, drifting nights and days, over oceans and continents, and above convective and non-convective regions. The fraction of time during which the flow is found to be turbulent in the lower stratosphere appears to be quite small, ranging from 3.3 to 6.6%. Because of the lack of sampling bias, such a

fraction of time can be interpreted as a fraction of space. The probability of occurrence of instabilities is far from being uniform in time (or space). Periods of several days are frequently observed during which the atmosphere is stable, i.e. without any instability. On the other hand, during certain periods, the frequency of instabilities appears to be quite high. At first sight, these differences could be attributable to the underlying deep convection, as observed for example when the balloons are flying over the maritime continent. Yet, this remains very preliminary conclusions that need to be substantiated in future work.

*Data availability.* The balloon-borne TSEN and RACHuTS data used in this study were collected as part of Strateole-2, which is sponsored by CNES, CNRS/INSU, NSF, and ESA. The Strateole-2 data set is available at https://data.ipsl.fr/catalog/strateole2/.

**Appendix A:  Estimation of the uncorrelated noise**

The measured signal is assumed to contain an uncorrelated and centered noise contribution. This noise level is estimated on short data segments ($\sim$ 20 data samples), the useful signal being described by a polynomial fit of degree $n$. The time series of

quantity X reads

$$\underline{X_i} = X_i + \xi_i \quad (1 \leq i \leq n) \tag{A1}$$

where $\underline{X_i}$ is the measured signal, and $\xi_i$ an uncorrelated noise of variance $\sigma_\xi$. The measured first increment reads:

$$\delta\underline{X_i} = X_{i+1} - X_i + \xi_{i+1} - \xi_i \tag{A2}$$

If $X_i$ is constant, i.e. $X_{i+1} - X_i = 0$, the variance of $\delta\underline{X_i}$ reduces to

$$\mathrm{Var}[\delta\underline{X_i}] = 2\,\mathrm{Var}[\xi] = 2\sigma_\xi^2 \tag{A3}$$

The measured second increment reads:

$$\delta^2\underline{X_i} = \delta\underline{X_{i+1}} - \delta\underline{X_i} = X_{i+2} - 2X_{i+1} + X_i + \xi_{i+2} - 2\xi_{i+1} + \xi_i \tag{A4}$$

If $X_i$ varies according a linear trend, i.e $X_{i+2} - 2X_{i+1} + X_i = 0$, the variance of $\delta^2\underline{X_i}$ reduces to:

$$\mathrm{Var}[\delta^2\underline{X_i}] = \mathrm{Var}[\xi] + 4\,\mathrm{Var}[\xi] + \mathrm{Var}[\xi] = \sum_{k=0}^{2} \binom{2}{k}^2 \sigma_\xi^2 = 6\sigma_\xi^2 \tag{A5}$$

The measured $n$-th increment reads:

$$\delta^n \underline{X_i} = \delta^{n-1}\underline{X_{i+1}} - \delta^{n-1}\underline{X_i} = \sum_{k=0}^{n}(-1)^{n-k}\binom{n}{k}X_{i+k} + \sum_{k=0}^{n}(-1)^{n-k}\binom{n}{k}\xi_{i+k} \tag{A6}$$

If $X_i$ is described by a polynomial of degree $n-1$, the first term on the right-hand side of Eq. (A6) cancels, and the variance of $\delta^n \underline{X_i}$ reduces to:

$$\mathrm{Var}[\delta^n \underline{X_i}] = \sum_{k=0}^{n}\binom{n}{k}^2 \sigma_\xi^2 \tag{A7}$$

Increasing the order of differentiation enhances the relative contribution of the uncorrelated signal in the time series. After several differentiations, the variance of the differentiated time series is expected to converge to the weighted variance of the uncorrelated noise.

## Appendix B: Impact of instrumental noise on the correlation coefficients

The Pearson or Spearman correlation coefficients $\rho_P$ and $\rho_S$ will be reduced because of uncorrelated noise in the timeseries of $\delta z_B$ and $\delta\theta$. A simplistic model may help to illustrate this assertion. Let us note $\underline{\theta}$ the measured potential temperature and $\underline{z_B}$ the measured altitude of the balloon. Assume that $\underline{\theta}$ can be analyzed as $\underline{\theta} = \theta + \eta$, where $\theta$ is the real value and $\eta$ is a centered random noise. Similarly, $\underline{z_B} = z_B + \zeta$. The variance of the measured increments $\delta\underline{X}$ ($X = \theta$ or $z_B$) reads: $\mathrm{Var}[\delta\underline{X}] = \mathrm{Var}[\delta X] + 2\sigma_X^2$ where $\sigma_X$ is the standard deviation of the random noise on $X$. Also $\mathrm{Cov}[\delta\underline{\theta}, \delta\underline{z_B}] = \mathrm{Cov}[\delta\theta, \delta z_B]$. The expectation of the Pearson correlation coefficient $\rho_P[\delta\underline{\theta}, \delta\underline{z_B}] = \mathrm{Cov}[\delta\underline{\theta}, \delta\underline{z_B}]/(\mathrm{Var}[\delta\underline{\theta}]\,\mathrm{Var}[\delta\underline{z_B}])^{1/2}$ reads:

$$\rho_P[\delta\underline{\theta}, \delta\underline{z_B}] = \frac{\rho_P[\delta\theta, \delta z_B]}{\left(1 + \dfrac{2\sigma_\zeta^2}{\mathrm{Var}[\delta z_B]} + \dfrac{2\sigma_\theta^2}{\mathrm{Var}[\delta\theta]} + \dfrac{4\sigma_\zeta^2\sigma_\theta^2}{\mathrm{Var}[\delta\theta]\,\mathrm{Var}[\delta z_B]}\right)^{1/2}} \tag{B1}$$

where $\sigma_\theta^2$ and $\sigma_\zeta^2$ are the variances of the noises on $\theta$ and $z_B$, respectively. The correlation coefficients are thus expected to be significantly reduced due to the instrumental noise. Yet, they will retain the same sign as the correlation coefficients without instrumental noise.

*Author contributions.* RW developed the data processing method and wrote the major part of the manuscript. CP participated in conditioning and processing the data. AP processed the RACHuTS data. AH is PI of Strateole-2, he pre-processed the raw data, making them available. AP and AH contributed importantly in exchanging extensively about the physics of super-pressure balloon borne measurements. All authors (RW, CP, AP, AH, MC and RP) participated in numerous discussions to elaborate and improve the overall understanding of the physics of measurements, and all co-authors contributed significantly to the writing of the paper.

*Competing interests.* The authors declare no conflicts of interest relevant to this study.

*Acknowledgements.*  The authors acknowledge the support of Agence National de la Recherche through project TuRTLES (grant agreement ANR-21-CE01-0016-01). Clara Pitois is founded through a Sorbonne-Universite doctoral fellowship.

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



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
