# Peer review of "Detection of Turbulence from Temperature, Pressure and Position Measurements Under Superpressure Balloons."

_Atmospheric Measurement Techniques, 2022_

## Referee Comment (RC1)

**Peer-Review of "Detection of Turbulence from Temperature, Pressure and Position Measurements Under Superpressure Balloons."**

August 2022

**1 Summary of the Content**

The manuscript of Wilson and coauthors establishes a new way of assessing whether a super-pressure balloon flew through calm or turbulent air. In short, this is done by comparing the balloon's vertical oscillations around its equilibrium height to the vertical temperature gradient. If there is a non-positive correlation between altitude and potential temperature increment, this is seen as the result of turbulent mixing. Hence, the turbulence flag is set *TRUE*. Furthermore, this approach is compared to the classical Richardson criterion, where the flow is expected to be turbulent for $Ri < 0.25$. Both methods are used on several superpressure balloon flights from the Strateole-2 CO campaign, each method using two slightly different computational methods (different correlation estimators and different linear fitting methods).

**2 Overall Feedback**

First I would like to thank the authors for sharing their method. Especially, I liked their approach of explaining their ideas in simple terms the first place, followed by a rigorous mathematical description thereafter. As far as my knowledge goes, the method is new and I really appreciate having another tool to retrieve turbulence from standard instrumentation on a superpressure balloon, even though the strength of turbulence cannot be quantified. Furthermore, I really like the careful consideration of instrumental noise in the approach. Their referencing in general is appropriate. Their use of the English language is easy to understand and correct as far as I can judge. The only comments I have on this are listed in Section 4 and Section 5 of this review.

Even though I very much like the general approach of the method and presentation, I think the manuscript will benefit from a *Discussion* section after "Results" and before "Summary and Concluding Remarks". This could especially enhance the discussion of the new correlation method in comparison with the more established Richardson method. I guess that this will foster confidence in the results from the new method, as it cannot be compared to a quantitative turbulence retrieval using the given data set.

Overall, I think that the approach shown by the authors is a very valuable contribution to the field of turbulence retrievals. It is well presented and absolutely fitting the scope of AMT. I recommend publication after minor revision. In Section 3 I list some points that according to my perception would complement the work of the authors especially in terms of discussing their results.

**3 Main Comments**

**3.1 Similarity of results from correlation method and Richardson method**

In ll. 390-391 you state that "The percentage of similar turbulence detection with these four indices ranges from 97.9% ($r_S$ vs. $Ri_{LSF}$) to 99.07% ($Ri_{LSF}$ vs. $Ri_{TSF}$). The different turbulent indices are therefore very consistent." Unfortunately, I do not immediately see this conclusion. If we for example consider the correlation method using $r_S$ vs. the Richardson method using $Ri_{TSF}$, Table 6 reveals 98.59% of identical turbulence detections for flight 02_STR2. Table 5, however, states that only 4.4% (3.7%) of the respective flight was found to be turbulent. *98.59% of identical turbulence detections* sound like a high number in the first place, but it could still mean that up to one third of the turbulence detections using one method are not seen using the other.

Could you maybe discuss this a little further? In my opinion it might be helpful to use the Richardson method ($Ri_{TSF}$) as the "established method" and then compare how specific and how sensitive the new correlation method is (e.g. $r_S$). You could even use a bar diagram, stating "correct positive", "correct negative", "false positive" and "false negative". Does that work for you?

**3.2 High occurrence rate of negative Richardson numbers**

I was pretty surprised by your manuscript finding a super adiabatic temperature gradient in 80% of all turbulent cases (l. 367), corresponding to a high occurrence rate of negative Richardson numbers for turbulent cases during flight 2 (Fig. 13). However, in Figure 11 the probability density for $N^2 \leq 0$ is given as zero. If $Ri$ is calculated from the equation in line 213 in both cases on a 1 hr average, I do not fully understand how both results can occur at the same time. But maybe this is a misunderstanding on my side.

According to my knowledge, most of the turbulence is described to occur for $N^2 > 0$ in the literature (e.g. Ko et al., 2019).

As said above, this may well be a misunderstanding on my side. If on the other hand side most of the turbulence in your measurements occurs for $Ri \leq 0$ indeed, could you maybe bring that up in the discussion? I would find that very interesting.

**3.3 Possible influences by warm downwash from the balloon**

On ll. 139-143 you state that flight 03_TTL3 shows a higher noise influence due to the temperature sensor being located closer to other parts of the gondola. Given the amplitude of the vertical balloon oscillation of up to 100 m (l. 193), the oscillation could also bring the sensors into the much larger wake of the balloon. This would then cause longer lasting warm spikes that may not easily be recognized as an instrumental effect. This is well known for ascending balloons (e.g. Gaffen, 1994; Kräuchi et al., 2016; Söder et al., 2019; Tiefenau and Gebbeken, 1989). It might be helpful discuss this possibility in a very short manor and to include a sketch of the flight train so that the reader can more easily assess the situation. As said, maybe this doesn't apply to your measurements, but it might be easier to see with the help of a sketch or similar.

**3.4 Nature of turbulence that can be detected by the correlation method**

I think that it could be beneficial for the reader to discuss the type of turbulence that can be detected with your knew method a little further in the discussion.
In the introduction, you state by various citations that many turbulence encounters in the equatorial UTLS are driven by KHIs (ll. 38-44). Your correlation method relies on the instrument being immersed in an iso-thermal layer (equal potential temperature). For a standard KHI, this is expected in the center region, but not at the edges, where at the beginning of the turbulence

development also strong viscous dissipation $\varepsilon$ takes place. Regarding the evolution of KHIs I find Fritts et al. (2003) and Werne and Fritts (1999) very valuable contributions, one of which you already cited. However, this point is not a *must* from my perspective, but rather a *nice to have*.

**4 Minor Comments**

ll. 112-113:  it could become more clear by saying "allows to obtain vertical temperature profiles of 2 km length below the balloon" instead of "allows to obtain 2-km long temperature profiles below the balloon"

ll. 118-119:  maybe rather "taken into account in order to assess whether" instead of "taken into account to assess that"

l. 237:  rather "is one of the few techniques" instead of "is one of the sole technique"

l. 245:  rather "expectation that is estimated by" instead of "expectation, is estimated by"

l.456:  maybe rather "Yet, they will retain the same sign as the correlation coefficients" instead of "Yet, they will remain of the same sign than the correlation coefficients"

**5 Typos**

l. 16:  probably "true *or* false" instead of 'true of false"

l. 34:  "radars" instead of "radar"

l. 102:  delete surplus "the"

l. 108:  "will for instance stand for" instead of "will for instance stands for"

l. 124:  either "signal spectra exhibit" or "signal spectrum exhibits"

l. 148:  "night/day" instead of "nigh/day"

l. 299:  should be "for each *of* the eight flights"

l. 350:  "red" should be "orange"

l. 361:  missing full-stop after "Figure 10"

l. 445:  "differentiations" instead of "differentiation"

**References**

Fritts, D., Bizon, C., Werne, J., and Meyer, C.: Layering accompanying turbulence generation due to shear instability and gravity-wave breaking, Journal of Geophysical Research: Atmospheres (1984–2012), 108, 2003.

Gaffen, D. J.: Temporal inhomogeneities in radiosonde temperature records, Journal of Geophysical Research: Atmospheres, 99, 3667–3676, doi:10.1029/93JD03179, 1994.

Ko, H.-C., Chun, H.-Y., Wilson, R., and Geller, M. A.: Characteristics of Atmospheric Turbulence Retrieved From High Vertical-Resolution Radiosonde Data in the United States, Journal of Geophysical Research: Atmospheres, 124, 7553–7579, doi:10.1029/2019JD030287, URL https://agupubs.onlinelibrary.wiley.com/doi/abs/10.1029/2019JD030287, 2019.

Kräuchi, A., Philipona, R., Romanens, G., Hurst, D. F., Hall, E. G., and Jordan, A. F.: Controlled weather balloon ascents and descents for atmospheric research and climate monitoring, Atmospheric Measurement Techniques, 9, 929–938, doi:10.5194/amt-9-929-2016, 2016.

Söder, J., Gerding, M., Schneider, A., Dörnbrack, A., Wilms, H., Wagner, J., and Lübken, F.-J.: Evaluation of wake influence on high-resolution balloon-sonde measurements, Atmospheric Measurement Techniques, 12, 4191–4210, doi:10.5194/amt-12-4191-2019, URL https://www.atmos-meas-tech.net/12/4191/2019/, 2019.

Tiefenau, H. K. E. and Gebbeken, A.: Influence of Meteorological Balloons on Temperature Measurements with Radiosondes: Nighttime Cooling and Daylight Heating, Journal of Atmospheric and Oceanic Technology, 6, 36–42, doi:10.1175/1520-0426(1989)006<0036:IOMBOT>2.0.CO;2, 1989.

Werne, J. and Fritts, D. C.: Stratified shear turbulence: Evolution and statistics, Geophysical Research Letters, 26, 439–442, doi:10.1029/1999GL900022, URL https://agupubs.onlinelibrary.wiley.com/doi/abs/10.1029/1999GL900022, 1999.

---

## Referee Comment (RC2)

**Review of AMT-2022-176**

The manuscript "Detection of Turbulence from Temperature, Pressure and Position Measurements Under Superpressure Balloons" introduces a new methodology to derive turbulence from super-pressure balloon observations. This has not been done before, which makes this paper very relevant and exciting. This new methodology will open exciting new possibilities of turbulence studies in an atmospheric region where otherwise sparse observation (if any) are available. The method seems sound to me and its description is understandable. However, I would recommend major revision due to three reasons:

- 1. There is no discussion on why spectral analysis methods cannot be used in this context and if it will be possible to make a statement about the intensity of the turbulence events.
- 2. I'm missing some at least high-level discussion about turbulence occurrence. I'm aware that the purpose of this paper is to introduce a new methodology and more in depth analysis will follow at a later point. But the authors already state that most of the turbulence is detected close to convection, however no proof is provided. Also, it would be rather straight forward to contrast the STR and TTL balloons e.g. was there more turbulence detected by TTL balloons?
- 3. The manuscript is full of typos, some examples are: line 20: than -> that; line 55: probaility-> probability; line 60: appears -> appear, line 100: the the -> the; line 100: in -> of; line 145: nigh-> night; line 200: smaller smaller-> smaller.... This is just to name a few, there are many more typos in the manuscript. Therefore, I highly encourage the authors to check the spelling.

Minor remarks

- Line 10: I thought the amplitude of the vertical balloon displacement is more on the order of 30m than 15m. Why is 15m cited here?
- Line 30: what about vertical transport through gravity waves?
- Line 75: "... obtained under super pressure balloons": do you mean "obtained with super pressure balloons"?
- Line 115: why did you degrade the RACHuTS measurements to 30m?
- Line 125f: "However, a rough estimate of the noise level has been evaluated...". I don't
  understand the tables and the noise calculation. How was this evaluated? Also the table
  states that it shows the standard deviation, while in the text the avaerage of the 10%
  smallest variances of the 6th order increments is mentioned. How does this fit together?
- Eq 1: what is pi?
- Line 195: the amplitudes range up to 100m, I'm wondering if some of the large amplitudes are due to depressurization events. How did you actually handle balloon depressurization events?
- Line 220: What are the corresponding spatial scales?
- Fig. 5: don't fully understand this figure: Shouldn't the red line indicating Theta2 be at the same level for both times t1 nd t1+30s? Maybe leave it out at t1?
- Fig. 6: how do these spectra compare to other flights?

- Line 270: Typos/grammar: this throughout the manuscript that singular and plurals are mixed up. Either one coefficient is smaller or coefficients are smaller... please also check the whole manuscript for these mixes.
- Fig. 7: my copy didn't contain green lines in the bottom panel.
- Line 332: "... the choice of the time series...": do you maybe mean "the choice of parameter.."?
- Line 335: Could you show that the two methods detect the same turbulent periods in one single figure to make it really obvious. Plot them above one another... The histograms look very different between these two methods. What does that imply for the turbulence statistics?
- Line 374: Could you highlight the percentages in the figure to provide more guidance?
- Line 387: What about these differences? Are they significant? What would these differences imply?
- Fig. 13: what does the thick black line show in the bottom panel? Is it Ri=0.25?
- Appendix A:
  - A2: what is Xi? Should it maybe read Xi?
  - A5/A6: what is c2 (A5) or cn (A6)?

---

## Author Response (AR1)

**AC#1**

We thank the reviewer for his careful reading of the article. His constructive comments should undoubtedly contribute to improving the paper. As suggested, we have added a "discussion" section to include some details on the comparison between the turbulence indexes, and also to show the inhomogeneity of the turbulence detections according to the position of the balloons.

Follows a point-by-point response to the reviewer's remarks and comments.

**Main comments**

**3.1 **Similarity of results from correlation method and Richardson method.**

The reviewer's comment is quite relevant. The fact that the diagnoses of the flow state, laminar or turbulent, are identical in 97 or 98% of the cases does not mean that the diagnoses of the turbulent cases alone are identical at that level. In fact, the detections are most consistent when the vertical stratification is high. In such cases, there is very little disagreement between the methods because the correlation levels, or Richardson numbers, are high.

However, the situation is quite different if we consider only the diagnoses of turbulent flows (between 3 and 6% of cases in the average). For these cases, the differences can reach a factor of two. Thus, for flight 7 (07_STR2), the percentages of detection of turbulent sequences vary from 3.3% (Ri_TSF) to 6.3% (r_P). We believe that these differences result mainly from the fact that the threshold values, zero correlation or Ri = 0.25, correspond to the tails of the distributions of these estimates (Figs. 9 and 13 of the paper). When the atmosphere is weakly stratified, threshold effects are likely to be important, leading to important differences in the diagnosis of the flow conditions. Also, the differences between the Ri and correlation methods can be partly due to the thresholds values of the hypothesis tests of a null correlation (i.e. choice of a confidence interval for the null correlation).

As suggested by the referee, we compared the detections by the 4 methods taking $Ri_{TSF}$ as a reference indicator. We have considered the four possibilities for each of the three estimators: correct-turbulent and false-turbulent, correct-laminar and false-laminar.

*Table 1: Percentages of true (identical) and false detection of the various methods compared to the Ri_LSF method.*

| $Ri_{TSF}$ | Turbulent | | Laminar | |
|---|---|---|---|---|
| | True | False | True | False |
| $Ri_{LSF}$ | 85.4 | 14.6 | 99.2 | 0.8 |
| Pearson Corr | 76 | 24 | 99.1 | 0.9 |
| Spearman Corr | 83.2 | 16.8 | 99.5 | 0.5 |

The table shows the percentages of correct (i.e. identical) and incorrect diagnoses by the $Ri_{LSF}$ and correlation methods compared to the $Ri_{TSF}$ detections. The bar chart below shows the same thing in graphic form. It can be seen that the diagnoses are identical in more than 99% of the cases if the flow is detected as laminar. On the other hand, the diagnosis are identical for about 80% of the cases if the flow is detected as turbulent. We attribute these poorer performances to the fact that the critical thresholds, Ri = 0.25 and correlation = 0, belong to the tails of the distributions of the statistics and that the edge effects are more important for these rare events.

[Figure]

Figure 1: *Percentages of true and false detections for the turbulent (T) and lamimar (L) episodes compared to the $Ri_{TSF}$ method. The three methods $Ri_{LSF}, r_P, rS$ are compared.*

We have added a paragraph and a figure in the article to clarify this fact.

**3.2 High occurrence rate of negative Richardson number**

*Ri* and $N^2$ time series for flight 2 (02_STR2) are shown in figure 11 and 13 . The probability for Ri ( $N^2$ ) to be negative is not zero since occurrences of negative values are visible in the time series. For the considered flight, the occurrence frequency of negative $N^2$ is 3.4% (from the Theil-Sen regression performed on $T_C$ and $Z_P$ ). Such negative Ri ( $N^2$ ) can result from both the dispersion of the temperature gradients estimates or the occurrences of episodes of unstable stratification. Note that we have corrected the histogram of $N^2$ in Figure 11. They were not plotted correctly in the original version of the paper since only positive classes were defined. Negative occurrences are now visible.

In the present study, negative estimates of *Ri* (or $N^2$ ) may be due to the precision of the estimates of the temperature gradients (scattered around a value close to -10°/km in case of quasi neutral stratification). Temperature gradients at the balloon flight level are estimated from the covariance of increments of temperature and displacements, these covariances being computed over one-hour time segments. Assuming neutral stratification, the covariances are expected to be scattered around 0, implying some negative estimates of $N^2$ .

However, unstable stratifications ( $N^2 < 0$ ) seem to occur in the lower stratosphere since they have been reported in the literature. For instance, detection of turbulence by the Thorpe method from insitu measurements is based on observations of, $\partial\theta/\partial z < 0$, i.e. $N^2 < 0$ (Thorpe, 1977). The probability of occurrence of such unstable layers likely depends on the vertical resolution of the profiles (see for instance Wilson et al., 2011, Geller et al., 2021) but it not zero. It is exact that for Kelvin-Helmholtz instabilities, turbulence is expected to be triggered for $0 < Ri < 1/4$, i.e. for $N^2 > 0$, but once it is developed, the stratification can become almost neutral ($N^2 \approx 0$), or even unstable ($N^2 < 0$), as a result of stirring and mixing. Therefore, it is plausible that the occurrences of such unstable episodes may also contribute to negative values for $Ri$ ($N^2$) based on covariances calculated on one-hour time segments.

Estimates of $Ri$, or $N^2$, from radiosondes, when applying the Thorpe's method, are made from the sorted potential temperature profiles – anywhere increasing with altitude - and therefore they cannot be negative. However, the measured profiles show decreasing potential temperature with altitude in some places (i.e. the stratification is unstable and $N^2 < 0$). This is at the base of the Thorpe detection method.

3.3 **Possible influences by warm downwash from the balloon**

The referee's remark is quite relevant. Indeed, due to the vertical oscillations of the balloon, the T-sensors are possibly in the wakes of the balloon or of the flight chain. Notice that we expect the balloon wake to be warm during daytime and cold during nighttime, the balloons being cooler than the ambient air during nighttime.

The diameter of the balloons is either 11 m (TTL) or 13 m (STR). The temperature sensors are located 27 m below the balloon base (except for TTL3 flight) and 15 m below the EUROS gondola. On all but TTL3 flights, the T sensors are located 7 m below the last gondola in the flight chain. Flight 03_TTL3, carrying the RACHuTS system, is an exception since the temperature sensors are located 30 cm away from the EUROS gondola.

The probability of the T-sensors being in the wake of the balloon or gondolas is clearly non-zero. If there is no horizontal wind shear, the T-sensors should enter the wake of the balloon as soon as they enter the area in which the balloon is oscillating (about 30 m wide). Taking into account the distance between the balloon and the T-sensors (27 m), the T-sensors can enter the balloon's wake only if the amplitude of the balloon oscillations is larger than ~13.5 m (27 m peak-to-peak), that is for slightly more than 50% of the time (the median value for amplitudes is 15 m). Anyway, the T measurements could still be perturbed by the wake of the flight chain (gondola(s), parachute, wires). The only case where the T-sensors should not enter the wakes is when the wind shear is sufficiently large (about 5 m/s/km).

The issue of the possible impact of wakes was considered during this study. Indeed we calculated the statistics, vertical gradients and correlations, considering only the phases when the balloon descends. During these phases, the temperature sensors (which are located at the lower end of the flight chain) sample the "fresh" air if a minimum shear exists. The figure below shows the time series of Pearson/Spearman correlations for the flight presented in the article (Fig. 9) but considering only the phases when the balloon descends. The resulting time series are noisier since we only consider about half of the samples. However, both time series of correlations and temperature gradients have similar characteristics to those calculated when considering all samples, showing the same succession of stable and unstable periods. We therefore conclude that the impact of the wake during the ascending motions does not affect significantly the estimated correlations

and covariances and because of the increase of noise we choose to consider all the samples. We did not mention it in the initial version of the article because we did not observe an important impact. There is now a paragraph in the "discussion" section about that point.

[Figure]

*Figure 2: Time series and histograms of correlations but considering only the descending phases of the SPB.*

**3.4 **Nature of turbulence than can be detected by the correlation method**

As the reviewer recalls, numerical simulations indicate that large temperature gradients are expected at the edges of turbulent layers (Fritts et al., 2003; Werne and Fritts, 1999). These strong gradients are also commonly observed from radiosonde profiles when turbulence detection is performed by the Thorpe method. It is clear that the sampling by the balloons, drifting within an air mass and not cutting vertically through it as a radiosonde, will not allow to identify such temperature gradients at the edge of turbulent regions. Only the central part of the turbulent region, in which stratification is almost zero, can be detected. We added a few sentences in the document to clarify this fact.

**Minor comments**

We warmly thank the reviewer for his suggestions (which we all followed) and for pointing out the typos, which were corrected.

**AC#2**

We thank the referee for its constructive remarks which should undoubtedly contribute to improve the present paper.

Follows a point-by-point response to the referee's remarks.

**Major remarks**

**1.** The sampling period of measurements is 30 s, i.e. $f_{Nyquist} = 1/60$ Hz. The oscillation periods of the balloon vary around 220 s, i.e. $f_{NBO} \approx f_{Nyquist}/3.5$ . We found that the variability of measured quantities in this narrow frequency domain is dramatically affected by balloon oscillations (see the *w* spectrum of Fig.6). This conclusion is given in lines 200-204 and 401-404. In fact, our first attempt to detect turbulence was to look for an excess of variance or a spectral signature associated with fluctuations of the measured parameters in the $f_{NBO} - F_{Nyquist}$ frequency range. This method was unsuccessful. The main reason, we think, is that the high-frequency fluctuations are dominated by the balloon's natural oscillations. Therefore, we turned to statistical methods to detect the effects of turbulence from sensors that do not allow to directly measure turbulence.

**2.** As noted by the reviewer, the purpose of this paper is not to describe geophysical results. We indicated however that the detection is not uniform around the globe but seems to depend on the position. The following figure showing the occurrence frequency of the turbulence detection supports this statement. The variability according to the position appears to be very large, ranging from zero to about 25%. However, we think that the interpretation of these results is beyond the scope of the article and we do not think to publish it here. A PhD thesis by one of the co-authors (CP) is in progress on the geophysical exploitation of strateole-2 data.

We furthermore think that the turbulence we observe in the stratosphere is primarily due to wave breaking. As commonly accepted, deep convection is a major source of waves in the tropics, and those waves propagate vertically in the atmosphere. We therefore do not expect much difference between the occurrences of turbulence in the TTL and STR balloon flights: those are only 2 km apart. On the other hand, we expect a significant longitudinal modulation.

From Table 5 indicating the fraction of time during which the flow is turbulent does not show any clear contrast between the TTL and STR balloons, but it is not conclusive as only two TTL balloons are reliable (excluding flight 3, very noisy).

[Figure]

*Figure a: Trajectory of flight 6 (STR1) of the C0 campaign (top) and percentage of turbulent detection by the Ri method (bottom).*

**3.** We thank the reviewer for pointing out the (too) many typos for which we apologize. The mentioned typos – and others - have been corrected.

**Minor remarks**

- **line 10**: The mode of the distribution of the vertical displacements associated with the oscillatory motions of the SPBs is ~ 15 m (see figure 4).

- **line 30**: Gravity waves transport vertically energy and momentum. As long as waves do not saturate, there is no impact on the dynamics or chemistry of the atmosphere. Heat, momentum and minor-constituent fluxes occur when wave saturation occurs, i.e. when waves break into turbulence.

- **line 75**: We wrote "under high pressure balloons because the measurements are carried out on board of a gondola and temperature sensors located under the balloons.

- **line 115**: we downgrade the RACHuTS measurement to 30 m 1) to improve the measurement precision, and 2) in order to reach similar vertical resolution for TSEN and RACHuTS, TSEN estimates being evaluated in layers of about 30 m thick (+/- 15 m) -.

- **line 125**: The instrumental noise level is quantified by the standard deviation of the high frequency fluctuations. This standard deviation is estimated from the variance of the nth order increments as described in Appendix 1. Because both the balloon motions and the instrumental noise contribute to the high-frequency variability, we consider only the smallest 10% of estimates to minimize the influence of the balloon motions.

- **Eq 1**:   $\pi \approx 3.14159\ldots$

- **line 195**: The referee's remark is quite right. Most large amplitude oscillations are associated with depressurization events. We have added a sentence to clarify this fact:"A few large amplitude

oscillations (>100 m) are observed. We found that they are most often associated with depressurization events. ".

We did not handle balloon depressurization events because they do not appear to have impact on turbulence detection (we do not detect turbulent events during depressurization events).

- **line 220**: Since the horizontal wind is typically of the order of 5 m/s, the horizontal spatial scale is typically 18 km. But this horizontal scale does not correspond to the spatial extension of the observed episodes (turbulent or laminar) since the balloon is advected by the flow, i.e. the observations are Lagrangian. Lagrangian observations of turbulent or laminar events are rather related to the lifetime of the events.

- **Fig 5**: The red line at $t_1$ indicates the potential temperature level that the balloon will reach at $t_1+30$ s. It is drawn to indicate schematically that the whole layer is vertically displaced in the 30 s time interval. Therefore, the observed $\theta$ changes are not simply proportional to the vertical $\theta$ gradient.

- **Fig 6**: The two spectra in Fig. 6 are representative of the power spectra of $w$ for all flights: almost flat up to the frequencies corresponding to the oscillatory motions of the balloons. Similar power spectra have already been published (Podglagen et al.).

- **line 270**: Thanks to the reviewer for pointing out the (too) many typos. We have corrected those mentioned, and others.

**Fig 7**: The colors of lines of the plot have been corrected

**line 332**: "the time series" → 'parameters'

**line 374**: The probabilities are indicated as the y-axis of the cumulative distribution function (bottom right). The percentages are deduced.

**line 387**: The differences about the turbulence detections from the different methods are discussed in the paragraph "Discussion", section 5.1 and are shown in the new figure 14.

The detection methods are based on statistics (two correlations and two regressions) applied on independent measurements. The percentages of turbulence differ by using different estimators or different parameters: the percentage differences can reach a factor of two (table 5, flight 07_STR2, 6.3% vs. 3.3%). These differences can be partly due to the thresholds values of the hypothesis tests of a null correlation (choice of a confidence interval). Likely more important is the fact that the thresholds values (null correlations or $Ri=0.25$) are associated with the tails of the distributions of these statistics (see the histograms of figures 9 and 13). The differences occur mainly when the atmosphere is close to a neutral stratification, i.e. when the estimates are close to the thresholds. When the atmosphere is somewhat stratified (about 95% of the time) the statistical estimators are very consistent, the agreement between the estimates being larger than 99% (Fig. 14 of the paper).

The following figure shows the turbulent periods detected by the four different methods for flight two. As stated in the paper, we privileged the combination of $T_C$ and $z_P$ as they appear to

[Figure]

*Figure b: Time series of the turbulence detection from four estimators.*

be less noisy than $T_S$ and $z_{GPS}$. Overall, the agreement is good, even differences are visible.

In short, the used statistics (correlations and Ri) allow to describe the state of stratification of the atmosphere in which the balloons float. When the atmosphere is close to a neutral state, all estimates, correlation or *Ri*, reach values belonging to the tail of their distributions, close to threshold values. Some differences in detecting the turbulent episodes occur due to threshold effects. We added a subsection about the comparison of the different indexes (.

Appendix A: The two typos in the equations have been corrected.

**AC#3**

We thank the reviewer for his careful reading of the article. His constructive comments should undoubtedly contribute to improving the paper.

Follows a point-by-point response to the referee's remarks.

- **The referee states that the title of the paper is misleading because we do not present turbulence data.** It is correct that we do not present turbulence measurements as the oscillating motions of balloons does not allow direct observations of turbulence. However, the paper presents methods for detecting turbulence based on estimates of the local stratification. The underlying assumption of such an approach is similar to that of Thorpe's detection (Thorpe, 1977) or to detection methods based on the Ri criterion: in case of unstable flow, i.e. when $Ri < ¼$ or $\theta_z \leq 0$, the flow is assumed to be turbulent. It is true that these conditions may be the cause of turbulence - in the initial phase of an instability - before turbulence develops. However, the statistics on which stratification is estimated, correlations or covariances, are calculated over one-hour time segments. It appears very unlikely that an unstable flow could persist for such a time without generating turbulence.. Therefore, we believe that the detection of neutral or even unstable stratification is, or has been recently, associated with turbulent episodes. We therefore think that the current title is appropriate.

- **About the lack of other evidence of turbulence**: our first attempt to detect turbulence was to look for high-frequency excesses of variance or a high-frequency spectral signature of the measured fluctuations. This method was unsuccessful. We believe this is because the high frequency variance is dominated by the natural oscillations of the balloon. The sampling period of measurements is 30 s, i.e. $f_{Nyquist} = 1/60$ Hz. The oscillation periods of the balloon vary around 220 s, i.e. $f_{NBO} \approx f_{Ny}/3.5$. We found that the variability of measured quantities in this narrow frequency domain is dramatically affected by balloon oscillations (see the *w* spectrum of Fig.6). This point is discussed in the paper (lines 199-203 and 401-404). Therefore, we turned to statistical methods to detect the effects of turbulence from sensors that do not measure turbulence directly.

  Despite the fact that we cannot directly measure the turbulence intensity from measurements under SP balloons, we show that the vertical temperature gradients estimated by statistical methods from TSEN measurements are consistent with the RACHuTS measurements. Also, observing $\omega_{NBO} < \omega_B < N$, as modeled by Podglagen et al. (2016, see the supplementary information) gives confidence in the estimates of the vertical gradients since $\omega_{NBO}$ and *N* depend on them (Eqs. 1 and 2). Therefore, we are confident about the estimates of vertical gradients and correlations revealing periods with little or no stratification of the flow.

- **About the lack of relationship between the times and places where turbulence occurs and the synoptic conditions**: the objective of the paper is clearly methodological and we did not investigate for the causes of turbulence occurrences. We have only mentioned the fact that the frequency of occurrence of turbulence is about 5% in the average and is not uniform around the globe. It appears to be greater over convective areas such as the maritime continent. Such an assertion is justified by the following figure which shows the positions of the turbulence detections (Ri method) as green dots for the 8 flights of the C0 campaign. At the present stage, this result is qualitative and need to be deepened. Following the suggestion of the reviewer we have included this figure in the new version of the paper. The exploitation of the Strateole-2 data set with the presented methods is the subject the PhD thesis of one of the co-authors (C.P.).

[Figure]

*Fig. 3.1 Positions of the turbulence detection for the height flights of the C0 campaign.*

- **About the assumption that layers with neutral stratification are turbulent**: There is no doubt that instability precedes the development of turbulence. We agree that this instability condition (Ri < 1/4) may be due to dynamic processes such as waves or shear, and not to turbulence. But such instability cannot persist and will necessarily generate turbulence. Therefore, if unstable, or neutral, stratification is observed, either turbulence is developed or it will develop. Since the statistics are based on one-hour segments, we favor the first hypothesis.

  We modified the sentence of line 205 to precise this point: "A consequence of turbulence is to restore stability from a preceding unstable state of the flow, which is achieved by locally mixing the fluid" and added a sentence: : "Note that neutral or even unstable stratification conditions may precede turbulence, but such conditions cannot persist and will cause turbulence."

- **About the impact of turbulence on vertical transport**: the question of the impact of turbulence on the vertical transport is the major motivation for turbulence study. This does not imply that we can answer this question with this article, which is only methodological. An estimate of vertical transport will have to rely on a physical model constrained by observations. The fact that we detect turbulence in the average for about 5% of the time does not allow us to quantify the vertical transport. The question of the impact of turbulence on transport is the central question of the PhD thesis of C.P..

- **About the turbulent fraction of the atmosphere**: the turbulent fraction in the UTLS is very poorly known. The few estimates from radiosondes or instrumented aircraft suggest values between 0 (no detection at all) and about 5%. It is difficult to compare these values because they are based on instruments with different sensitivities.

  Beyond an average value for the tropical UTLS, we found that the spatial distribution of turbulent events around the globe is far from uniform. The differences are very large since the turbulent fraction can be higher than 20% in some regions and almost zero in others. Interestingly, these inhomogeneities provide information on the mechanisms behind these turbulent episodes.

  In addition, the Lagrangian observations enabled by SPBs provide information on the lifetime of turbulent events. This parameter is crucial in some models (Alisse & Haynes) to quantify the turbulent transport.

- **line 60**: Indeed, there are many studies of turbulence from aircraft measurements. We added a reference to a recent work (Dörnbrack et al. 2022) with numerous references. However, to our knowledge, these studies do not involve tropical UTLS (with the exception of Podglajen et al., 2017, who used data obtained above &the ceiling of most research aircraft). Note that the current lack of turbulence measurements in that region is a major motivation for the present research (lines 33-35).

- **Table 1**: All balloons carry TSEN sensors. This has been specified in the legend of Table 1.

- **line 115**: Following the remark of the reviewer, the sentence has been modified: "… which we degraded to about 30 m in order to (1) improve the raw 1-m accuracy of the altitude measurements and (2) achieve similar vertical resolution from the TSEN and RACHuTS measurements (since the amplitude of the balloon oscillations is typically +/- 15 m)."

- **line 152**: The following figure (Fig. 3.2) shows the relative variations of volume of a Strateole-2 balloon during a two and a half month flight.

[Figure]

Fig. 3.2 Time series of the relative variations of the balloon volume during a flight.

The volume of the balloon varies by less than 3% peak to peak, the variations of diameter are thus lower than 1%. This precision has been added in the paper. The volume variations are mainly due to the diurnal cycle of the solar heating.

- **Fig. 8**: The large orange polygon extending to the left of -10 K/km represents the histogram of vertical temperature gradients for the only time intervals during which the flow is detected as turbulent. The green histogram (the color has been corrected) corresponds to the cases when the flow is stable.

- **line 395**: As suggested, we have removed the word "mainly" in the sentence.

- **line 404**: In the initial phase of this study, we looked for a high frequency signature as a proxy for turbulence, excess of variance or spectral signature. This method proved unsuccessful because the high frequency fluctuations ( $N/2\pi < f < 1/60$ Hz) are affected by the natural oscillations of the balloon.

However, it is planned for the next Strateole-2 campaign to perform fast (1s) velocity measurements with a sonic anemometer. This instrument, called VATA, is currently under development. These measurements should allow to directly measure turbulence.

- **line 424**: As suggested, we have included in the paper the plot of the positions of the turbulence detections (Ri method) for the 8 flights of the C0 campaign (Figure 3.1). The study to determine the origins of instabilities is ongoing.

---

## Referee Report (RR1)

**Peer-Review of revised manuscript "Detection of Turbulence from Temperature, Pressure and Position Measurements Under Superpressure Balloons."**

November 2022

**1 Overall Feedback**

As this is a review of a revised manuscript, I will skip the summary of the content.

First of all, I would like to thank the authors for the impressive amount of work they put into revising their manuscript. All points from my previous review have been extensively addressed. I think that the additions to the text are very valuable in putting their results into perspective. Furthermore, their discussion of possible technical issues enables the reader to get more confidence in the results that are presented in the manuscript. Concerning the occurrence of negative Richardson numbers: thank you for your clarifications and for pointing out that the histogram in Figure 11 has been corrected. My only concern was the high abundance of $Ri < 0$, which seems to have been a simple error.

Personally, I really like to thank the authors for taking on board the comments in such a meticulous way. This just makes the review process much more rewarding for the reviewer and fruitful for the scientific community.

My overall recommendation for the manuscript is to be published subject to some minor technical and typographical corrections and suggestions, which I will list below.

**2 Technical and Language Related Suggestions**

ll. 119-120:  This is a rather long sentence. Maybe split in two by replacing "$\sim$ 1 m, which we degraded" by "$\sim$ 1 m. We degraded this resolution".

ll. 222-224:  I find the last part of this sentence somewhat difficult to understand. Maybe replace "will not allow to identify as turbulent such temperature gradients." by "will not allow to identify turbulence from these sharp temperature gradients at the layer edges."

ll. 455-456:  Could you maybe add a reference here? I do not doubt your statement, it could just be interesting for those readers who are not that familiar with UTLS turbulence research.

ll. 458-459:  This statement could possibly also be strengthened by a reference, though I do not think it is a must. Fritts and Alexander (2003) would be a very generic choice from my point of view. Possibly you know one that is more specific?

l. 468:  Unfortunately, your last sentence of the of the paragraph is not immediately clear to me in terms of phrasing. What do you think of the following alternative: "A study examining processes that cause turbulence in the UTLS is in progress."?

**3 Typos**

l. 413: probably "wake" instead of "wakes"

l. 413: I assume "flight chain" instead of "chain flight"

ll. 425-426: rather "significantly affect" instead of "affect significantly"

l. 442: "threshold" instead of "thresholds"

l. 458: "is not zero" instead of "it not zero"

l. 442: "stratosphere" instead of "stratosphère"

l. 466: "Western Pacific" instead of "western Pacific"

- Throughout the text you use "indexes" as well as "indices". Maybe choose one of those options.

**References**

Fritts, D. and Alexander, J.: Gravity wave dynamics and effects in the middle atmosphere, Reviews of Geophysics, 41, 2003.

---

## Author Response (AR2)

**Answer to referee #1**

The authors warmly thank the referee for his positive comments. We sincerely think that the reviewer remarks contributed to significantly improve the content of the article and we are grateful for this.

And thank you for all the language suggestions that we all followed.

Line 455-456: we do not know any reference about the spatial distribution of turbulence in the tropical UTLS.

Also, thank you very much for pointing out the typos.

**Answer to referee #3**

The authors warmly thank the referee for his positive comments. We sincerely think that the reviewer remarks and comments contributed to significantly improve the content of the article and we are grateful for this.

**About the title of the paper**: it is exact that we "do not detect turbulence but layers which are conditioned to be turbulent". The paper describes methods for detecting turbulence occurences based on estimates of the local stratification. We therefore propose the following title: "Detection of Turbulence Occurrences from Temperature, Pressure and Position Measurements Under Superpressure Balloons"

Following the suggestion of the referee, **we have shortened the abstract** in removing few technical information.

Since we point out that only one study based on aircraft measurements provides information on turbulence in the tropical UTLS, we do not think it useful to add that few aircraft reach the altitude of interest and that of those few, not all are equipped with high quality turbulence sensors.

**About the fact that the oscillation frequency is larger than the BV frequency**, the answer is in Eq (2).
Expressing omega_NBO^2 as a function of N^2 :
omega_NBO^2 = 2/3N^2 + 10/21 g/H    (H = RaT/g ~ 6500 m)
omega_NBO^2 > N^2  => N^2 < 30/21 g/H ~ 22.5 e-4 s^-2.  Such a condition is met in almost all situations encountered.